# Agent-to-Sim: Learning Interactive Behavior Models from Casual Longitudinal Videos

**Gengshan Yang[1]**   **Andrea Bajcsy[2]**   **Shunsuke Saito[1]***   **Angjoo Kanazawa [3]***

[1]Codec Avatar Labs, Meta   [2]Carnegie Mellon University   [3]UC Berkeley

## Abstract

We present Agent-to-Sim (ATS), a framework for learning interactive behavior models of 3D agents from casual longitudinal video collections. Different from prior works that rely on marker-based tracking and multiview cameras, ATS learns natural behaviors of animal agents non-invasively through video observations recorded over a long time-span (*e.g.* a month) in a single environment. Modeling 3D behavior of an agent requires persistent 3D tracking (*e.g.*, knowing which point corresponds to which) over a long time period. To obtain such data, we develop a coarse-to-fine registration method that tracks the agent and the camera over time through a canonical 3D space, resulting in a complete and persistent spacetime 4D representation. We then train a generative model of agent behaviors using paired data of perception and motion of an agent queried from the 4D reconstruction. ATS enables real-to-sim transfer from video recordings of an agent to an interactive behavior simulator. We demonstrate results on animals given monocular RGBD videos captured by a smartphone. Project page: gengshan-y.github.io/agent2sim-www/.

## 1 Introduction

Consider an image on the right: where will the cat go and how will it move? Having seen cats interacting with the environment and people many times, we know that cats often go to the couch and follow humans around, but run away if people come too close. Our goal is to learn such a behavior model of physical agents from videos, just like humans can. This is a fundamental problem with practical application in content generation, VR/AR, robot planning in safety-critical scenarios, and behavior imitation from the real world (Park et al., 2023; Ettinger et al., 2021; Puig et al., 2023; Srivastava et al., 2022; Li et al., 2024; Schödl et al., 2000; Malle et al., 2023).

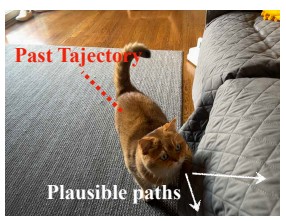

In a step towards building faithful models of agent behaviors, we present ATS (Agent-to-Sim), a framework for learning interactive behavior models of 3D agents observed over a long span of time in a single environment, as shown in Fig. 1. The benefits of such a setup is multitude: 1) It is accessible, unlike approaches that capture motion data in a controlled studio with multiple cameras (Mahmood et al., 2019; Joo et al., 2017; Hassan et al., 2021; Kim et al., 2024), our approach only requires a single smartphone; 2) It is natural – since the capture happens in the agent's everyday environment, it enables observing the full spectrum of natural behavior non-invasively; 3) Furthermore, it allows for longitudinal behavior capture, *e.g.*, one that happens over a span of a month, which helps capturing a wider variety of behaviors; 4) In addition, this setup enables modeling the interactions between the agents and the observer, *i.e.* the person taking the video.

While learning from casual longitudinal video observations has benefits, it also brings new challenges. Videos captured over time needs to be registered and reconstructed in a consistent manner. Earlier methods that reconstruct each video independently  (Song et al., 2023; Gao et al., 2022; Park et al., 2021) is not enough, as they do not solve correspondence across the videos. In this work, we tackle a more challenging scenario: building a *complete* and *persistent* 4D representation from orders of magnitude more data, *e.g.*, 20k frames of videos, and use them to learn behavior models of an agent.

---

*The last two authors equally mentored this project by both having babies.

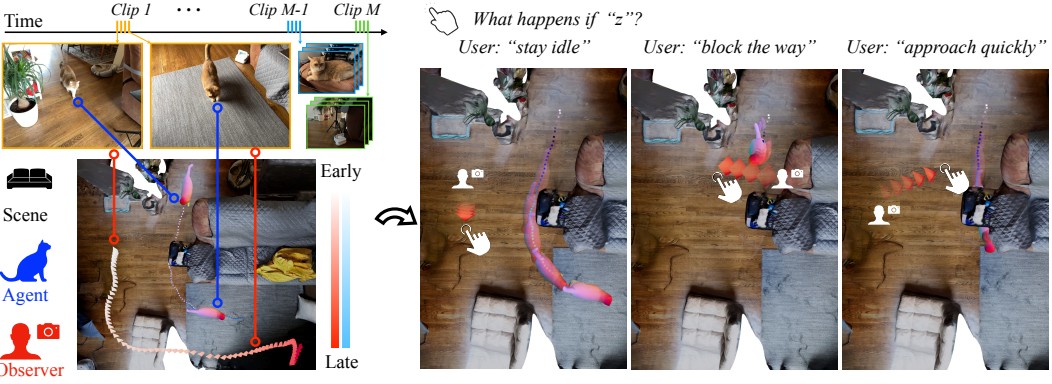

A) 4D Spacetime Reconstruction    B) Interactive Behavior Simulator

Figure 1: **Learning agent behavior from longitudinal casual video recordings.** We answer the following question: can we simulate the behavior of an agent, by learning from casually-captured videos of the *same* agent recorded across a long period of time (*e.g.*, a month)? A) We first reconstruct videos in 4D (3D & time), which includes the scene, the trajectory of the agent, and the trajectory of the observer (*i.e.*, camera held by the observer). Such individual 4D reconstructions are registered across time, resulting in a *complete* and *persistent* 4D representation. B) Then we learn a model of the agent for interactive behavior generation. The behavior model explicitly reasons about goals, paths, and full body movements conditioned on the agent's ego-perception and past trajectory. Such an agent representation allows generation of novel scenarios through conditioning. For example, conditioned on different observer trajectories, the cat agent chooses to walk to the carpet, stays still while quivering his tail, or hide under the tray stand.

To this end, we introduce a novel coarse-to-fine registration approach that re-purposes large image models, such as DiNO-v2 (Oquab et al., 2023), as neural localizers, which register the cameras with respect to canonical spaces of both the agent and the scene. While TotalRecon (Song et al., 2023) explored reconstructing both the agent and the scene from a single video, our approach enables reconstructing multiple videos into a complete and persistent 4D representation containing the agent, the scene, and the observer. Then, an interactive behavior model can be learned by querying paired ego-perception and motion data from such 4D representation.

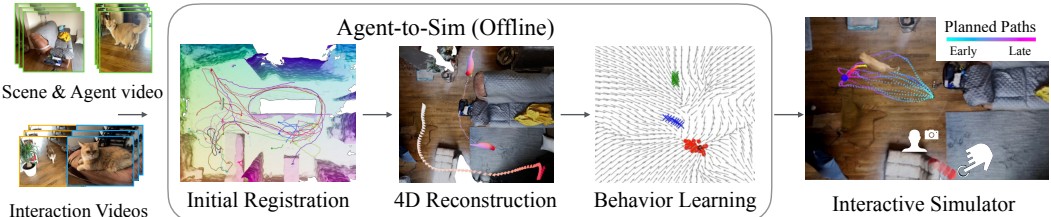

Scene & Agent video    Agent-to-Sim (Offline)    Planned Paths
Early    Late
Interaction Videos    Initial Registration    4D Reconstruction    Behavior Learning    Interactive Simulator

Figure 2: ATS takes videos of an agent and the scene and produces a interactive behavior simulator.

The resulting framework, as shown in Fig. 2, can simulate interactive behaviors like those described at the start: agents like pets that leap onto furniture, dart quickly across the room, timidly approach nearby users, and run away if approached too quickly. Our contributions are summarized as follows:

1. **4D from Video Collections.** We build persistent and complete 4D representations from a collection of casual videos, accounting for deformations of the agent, the observer, and changes of the scene across time, enabled by a coarse-to-fine registration method.

2. **Interactive Behavior Generation.** ATS learns behavior that is *interactive* to both the observer and 3D scene. We show results of generating plausible animal behaviors reactive to the observer's motion, and aware of the 3D scene.

3. **Agent-to-Sim (ATS) Framework.** We introduce a real-to-sim framework to learn simulators of interactive agent behavior from casually-captured videos. ATS learns natural agent behavior, and is scalable to diverse scenarios, such as animal behavior and casual events.

## 2 RELATED WORKS

**3D Agent Reconstruction from Monocular Videos.** Reconstructing time-varying 3D structures from monocular videos is challenging due to its under-constrained nature. Given a monocular video, there are multiple interpretations of the underlying 3D geometry, motion, appearance, and lighting (Szeliski & Kang, 1997). To deal with the ambiguities, prior methods often rely on category-specific body prior for *e.g.*, humans (Goel et al., 2023; Loper et al., 2015; Kocabas et al., 2020) and animals (Zuffi et al., 2017; 2018; 2019). However, parametric body models limits the degrees of freedom they can capture, and makes it difficult to reconstruct agents from arbitrary categories which do not have a pre-built body model, for example, mice and bunnies. Another line of works (Yang et al., 2022; Wu et al., 2021) avoid category-specific 3D priors and learns a flexible shape and deformation model of the agent (e.g., articulated bones) given pixel priors (*e.g.*, optical flow and object segmentation), which works for a wider range of categories including human, animals, and cars.

**World-space 3D Agent Reconstruction.** Beyond reconstructing the agents in the camera space, recent methods align reconstructed 3D humans to the world coordinate with the help of SLAM and visual odometry (Ye et al., 2023; Yuan et al., 2022; Kocabas et al., 2023). Sitcoms3D (Pavlakos et al., 2022) reconstructs both the scene and human parameters, while relying on shot changes to determine the scale of the scene. TotalRecon (Song et al., 2023) jointly optimizes the 3D agents, camera motion, and the 3D scene using compositional volume rendering, such that the motion of the agent can be decoupled from the camera motion and visualized from embodied viewpoints and fixed cameras in the world space. However, most of the method operates on a few hundreds of frames, and none of them can reconstruct a complete 4D scene while obtaining persistent 3D tracks over orders of magnitude more data (*e.g.*, 20k frames of videos). We develop a coarse-to-fine registration method to register the agent and the environment into a canonical 3D space, which allows us to leverage large-scale video collection to build agent behavior models.

**Behavior Prediction and Generation.** Behavior prediction has a long history, from simple physics-based models such as social forces (Helbing & Molnar, 1995; Alahi et al., 2016) to more sophisticated "planning-based" models that cast prediction as reward optimization (Kitani et al., 2012; Ziebart et al., 2009; Ma et al., 2017; Ziebart et al., 2008). With the advent of large-scale motion data, generative models have been used to express behavior multi-modality (Mangalam et al., 2021; Salzmann et al., 2020; Choi et al., 2021; Seff et al., 2023; Rhinehart et al., 2019). Specifically, diffusion models are used for behavior modeling for being easily controlled via additional signals such as cost functions (Jiang et al., 2023) or logical formulae (Zhong et al., 2023). However, to capture plausible behavior of agents, they require diverse data collected in-the-wild with associated scene context, *e.g.*, 3D map of the scene (Ettinger et al., 2021). Such data are often manually annotated at a bounding box level (Girase et al., 2021; Ettinger et al., 2021), which limits the scale and the level of details.

**3D Agent Motion Generation.** Beyond autonomous driving setup, existing works for human and animal motion generation (Tevet et al., 2022; Rempe et al., 2023; Xie et al., 2023; Shafir et al., 2023; Karunratanakul et al., 2023; Pi et al., 2023; Zhang et al., 2018; Starke et al., 2022; Ling et al., 2020; Fussell et al., 2021) have been primarily using simulated data (Cao et al., 2020; Van Den Berg et al., 2011) or motion capture data from multi-camera systems (Kim et al., 2024; Mahmood et al., 2019; Hassan et al., 2021; Luo et al., 2022). Such data provide high-quality body motion, but the interactions of the agents with the environment are either restricted to a flat ground, or a set of pre-defined furniture or objects (Hassan et al., 2023; Zhao et al., 2023; Lee & Joo, 2023; Zhang et al., 2023a; Menapace et al., 2024). Furthermore, the use of simulated data and motion capture data inherently limits the naturalness of the learned behavior, since agents often behave differently when being recorded in a capture studio compared to a natural environment. To bridge the gap, we develop 4D reconstruction methods to obtain high-quality trajectories of agents interacting with a natural environment, with a simple setup that can be achieved with a smartphone.

## 3 APPROACH

ATS learns behavior models of an agent in a 3D environment given RGBD videos. Sec. 3.1 describes our spacetime 4D representation that contains the agent, the scene, and the observer. We fit such 4D representation to a collection of videos in a coarse-to-fine manner, where the camera poses are initialized from data-driven methods and refined through differentiable rendering optimization

(Sec. 3.2). Given the 4D reconstruction, Sec. 3.3 trains an behavior model of the agent that is *interactive* to the scene and the observer. We provide a table of notations and modules in Tab. 6-7.

## 3.1 4D Representation: Agent, Scene, and Observer

Given multiple videos of an agent in their familiar environment, recorded over a long time horizon, our goal is to build a complete and persistent spacetime 4D reconstruction of the underlying world, including an evolving scene, a deformable agent, and a moving observer. The 4D representation is factored into time-independent canonical structures and time-varying components.

**Canonical Structure $\mathbf{T} = \{\mathbf{T_s}, \mathbf{T_a}\}$.** The canonical structure contains an agent neural field $\mathbf{T_a}$ and a scene neural field $\mathbf{T_s}$, following NeRF (Mildenhall et al., 2020). Scene properties, including densities $\rho$, colors $\mathbf{c}$, and semantic features $\psi$ are represented implicitly with MLPs. To query their values at any 3D location $\mathbf{X}$, we have

$$(\rho_s, \mathbf{c}_s, \psi_s) = \mathrm{MLP}_{scene}(\mathbf{X}, \boldsymbol{\beta}), \tag{1}$$

$$(\rho_a, \mathbf{c}_a, \psi_a) = \mathrm{MLP}_{agent}(\mathbf{X}). \tag{2}$$

The scene field takes in a learnable code $\boldsymbol{\beta}$ (Niemeyer & Geiger, 2021) per-video, which can represent scenes of slightly different appearance and layout (across videos) with a shared backbone. The MLPs are initialized with random weights and learned through inverse rendering.

**Time-varying Structure $\mathcal{D} = \{\boldsymbol{\xi}, \mathbf{G}, \mathbf{W}\}$.** The time-varying representation contains an observer and a deformable agent. The observer is represented by the camera pose $\boldsymbol{\xi}_t \in SE(3)$, defined as canonical-scene-to-camera transformations. We use BANMo (Yang et al., 2022) to represent the deformable agent, which contains a root pose $\mathbf{G}_t^0 \in SE(3)$, defined as canonical-agent-to-camera transformations, a set of articulated "bones" with time-varying centers and orientations $\{\mathbf{G}_t^b\}_{\{b=1,...,25\}}$, as well time-independent scales. Skinning weights $\mathbf{W}$ are defined as the probability of a point assigned to bones. Given the bone locations and scales, $\mathbf{W}$ is computed as the Mahalanobis distances between a point and bones, normalized by Softmax. A visual illustration can be found in Appendix Fig. 7. With this, any 3D location can be mapped between the canonical and the time $t$ space through blend-skinning (Magnenat et al., 1988),

$$\mathbf{X}_t = \mathbf{G}^a \mathbf{X} = \left(\sum_{b=1}^{B} \mathbf{W}^b \mathbf{G}_t^b\right) \mathbf{X}. \tag{3}$$

The bones are initialized uniformly on a sphere and optimized with inverse rendering.

**Inverse Rendering.** To learn the 4D representation $\{\mathbf{T}, \mathcal{D}\}$, we minimize the difference between the rendered pixel values and the observations using differentiable rendering, similar to NeRF (Mildenhall et al., 2020) training. Here we sample rays in the camera space at time $t$, use $\mathcal{D}$ to map them to the canonical spaces of the scene and the agent, and query values (*e.g.*, density, color, feature) from the canonical fields. Their values are composed in ray integration, similar to TotalRecon (Song et al., 2023). More details about volume rendering can be found in the appendix Sec. A.1. Next, we discuss how to set up the optimization such that it is well-behaved.

## 3.2 Optimization: Coarse-to-fine Multi-Video Registration

Due to small view overlaps between videos and the evolution of scenes (Sun et al., 2023), such as layout changes (*e.g.*, furniture get rearranged) and appearance changes (*e.g.*, table cloth gets replaced), it is challenging to find correspondence and align multiple videos to a global world coordinate (Sarlin et al., 2019). To solve this, we design a coarse-to-fine registration approach to robustify registration by relexing the requirement on precise correspondence. At its core, our method trains per-scene and per-agent camera pose regressors given template 3D assets, and uses them to align the observer and the agent to a global space for new videos of an evolving scene. Given the coarse alignment, inverse rendering is used to jointly optimize the 4D representation and adjust the cameras at a fine-level.

**Initialization: Neural Localization.** With the observation that large image models have good 3D and viewpoint awareness (El Banani et al., 2024), we adapt them for camera localization. We learn a

scene-specific neural localizer that directly regresses the camera pose of an image $\mathbf{I}$ with respect to the canonical representation,

$$\boldsymbol{\xi} = (\hat{\mathbf{R}}_0, \hat{\mathbf{T}}_0) = f_\theta(\boldsymbol{\psi}), \tag{4}$$

where $f_\theta$ is a ResNet-18 (He et al., 2016) and $\boldsymbol{\psi}$ is the DINOv2 (Oquab et al., 2023) feature of the input image. Geometric correspondence methods, such as DUSt3R (Wang et al., 2023), scale with $\mathcal{O}(N^2)$ memory and computation for $N$ images, which becomes infeasible for large-scale datasets (e.g., 10k images). In contrast, registering each image to a canonical representation reduces the cost to $\mathcal{O}(N)$, making it significantly more efficient and feasible to run at scale. To learn such a neural localizer, we capture a single template video of scene, and build a 3D map using off-the-shelf SfM tools, such as PolyCam. Given the template mesh, we synthesize paired data of images $\mathbf{I}$ and random camera poses $\mathbf{G}^* = (\mathbf{R}^*, \mathbf{t}^*)$ on the fly to train the neural localizer $f_\theta$,

$$\arg\min_{f_\theta} \sum_i \left( \| \log(\mathbf{R}_0^T \mathbf{R}^*) \| + \| \mathbf{t}_0 - \mathbf{t}^* \|_2^2 \right), \quad (\mathbf{R}_0, \mathbf{t}_0) = f_\theta(\boldsymbol{\psi}(\mathbf{I})), \tag{5}$$

where $\boldsymbol{\psi}(\cdot)$ is an off-the-shelf DINO-v2-small feature extractor. Geodesic distance (Huynh, 2009) is used for camera rotation, and $L_2$ error is used for translation. Rotations are represented as unit quaternions, where we force the real part to be positive to avoid the ambiguity in the representation. During training, we randomly sample camera poses, and apply image augmentations, including color jitter and masks to improve generalization. Similarly, we train a camera estimator for the agent. We first fit dynamic 3DGS (Luiten et al., 2024; Yang et al., 2023a) to a turnaround video of the agent with complete viewpoint coverage. The dynamic 3DGS is then used to generate synthetic data sampled from random viewpoints and different time instances to train a regressor that predicts root poses $\mathbf{G}^0$ from DINOv2 features. Visuals can be found in Fig. 8-9 of the appendix.

**Objective: Feature-metric Loss.** To refine the camera registration and learn the full 4D representation $\{\mathbf{T}, \mathcal{D}\}$, we use differentiable rendering to fit the model to images and DINO-v2 features of $M$ target videos $\{\mathbf{I}_i, \boldsymbol{\psi}_i\}_{i=\{1,...,M\}}$. We model 3D feature fields (Kobayashi et al., 2022) besides colors in our canonical NeRFs (Eq. 1-2), render them, and apply both photometric and featuremetric losses,

$$\min_{\mathbf{T}, \mathcal{D}} \sum_t \left( \| I_t - \mathcal{R}_I(t; \mathbf{T}, \mathcal{D}) \|_2^2 + \| \boldsymbol{\psi}_t - \mathcal{R}_{\boldsymbol{\psi}}(t; \mathbf{T}, \mathcal{D}) \|_2^2 \right) + L_{reg}(\mathbf{T}, \mathcal{D}), \tag{6}$$

where $\mathcal{R}(\cdot)$ is the renderer described in Sec 3.1. Compared to colors, feature descriptors from large pixel models (Oquab et al., 2023) are found more robust to appearance and viewpoint changes, which helps find coarse alignment across videos. We also apply a regularization term that includes eikonal loss, segmentation loss, flow loss and depth loss similar to TotalRecon (Song et al., 2023). During optimization, we use both the template videos and the target videos. The camera pose of the template scene video is set to the ground-truth from Polycam (not optimized). The observer (scene camera) and the agent's root pose in the target videos are initialized from the neural pose regressors.

**Scene Annealing.** To reconstruct a complete 3D scene when some videos are a partial capture (*e.g.* half of the room), we encourage the reconstructed scenes across videos to be similar. To do so, we randomly *swap* the code $\boldsymbol{\beta}$ of two videos during optimization, and gradually decrease the probability of applyig swaps from $\mathcal{P} = 1.0 \to 0.05$ over the course of optimization. This regularizes the model to share structures across all videos, but keeps video-specific details (Fig. 4).

### 3.3 INTERACTIVE BEHAVIOR GENERATION

Given the 4D representation, we extract a 3D feature volume of the scene $\boldsymbol{\Psi}$ and world-space trajectories of the observer $\boldsymbol{\xi}^w = \boldsymbol{\xi}^{-1}$ as well as the agent $\mathbf{G}^{0,w} = \boldsymbol{\xi}^w \mathbf{G}^0$, $\mathbf{G}^{b,w} = \mathbf{G}^{0,w}\{\mathbf{G}^b\}_{\{b=1,...,25\}}$, as shown in Fig. 6. Next, we learn an agent behavior model interactive with the world.

**Behavior Representation.** We represent the behavior of an agent in the world space over a horizon $T^* = 5.6$ seconds. This is achieved by a hierarchical model that generates goals (the final location of agent's root joint), path (trajectory of the root joint), and full body motion sequentially, as shown in Fig. 3. The body motion $\mathbf{G} \in \mathbb{R}^{6B \times T^*}$ is conditioned on path $\mathbf{P} \in \mathbb{R}^{3 \times T^*}$, which is further conditioned on the goal $\mathbf{Z} \in \mathbb{R}^3$. Such decomposition makes it easier to learn individual components compared to learning a joint model, as shown in Tab. 4 (a).

Figure 3: Pipeline for behavior generation. We encode egocentric information into a perception code $\omega$, conditioned on which we generate full body motion in a hierarchical fashion. We start by generating goals $\mathbf{Z}$, then paths $\mathbf{P}$ and finally body poses $\mathbf{G}$. Each node is represented by the gradient of its log distribution, trained with denoising objectives (Eq. 8). Given $\mathbf{G}$, the full body motion of an agent can be computed via blend skinning (Eq. 3). Gray arrows visualize the output of the denoising networks, which points to the direction to update the goal and path in the iterative denoising process.

**Goal Model.** We represent a multi-modal distribution of goals $\mathbf{Z} \in \mathbb{R}^3$ by its score function $s(\mathbf{Z}, \sigma) \in \mathbb{R}^3$ (Ho et al., 2020; Song et al., 2020). The score function is implemented as an MLP,

$$s(\mathbf{Z}; \sigma) = \mathrm{MLP}_{\theta_\mathbf{Z}}(\mathbf{Z}, \sigma), \tag{7}$$

trained by predicting the amount of noise $\epsilon$ added to the clean goal, given the corrupted goal $\mathbf{Z} + \epsilon$:

$$\arg\min_{\theta_\mathbf{Z}} \mathbb{E}_{\mathbf{Z}} \mathbb{E}_{\sigma \sim q(\sigma)} \mathbb{E}_{\epsilon \sim \mathcal{N}(\mathbf{0}, \sigma^2 \mathbf{I})} \left\| \mathrm{MLP}_{\theta_\mathbf{Z}}(\mathbf{Z} + \epsilon; \sigma) - \epsilon \right\|_2^2. \tag{8}$$

**Trajectory Models.** Similar to how we model goals, we represent paths with score function conditioned on goals, and represent body poses with score function conditioned on paths,

$$s(\mathbf{P}|\mathbf{Z}; \sigma) = \mathrm{ControlUNet}_{\theta_\mathbf{P}}(\mathbf{P}, \mathbf{Z}, \sigma), \tag{9}$$

$$s(\mathbf{G}|\mathbf{P}; \sigma) = \mathrm{ControlUNet}_{\theta_\mathbf{G}}(\mathbf{G}, \mathbf{P}, \sigma). \tag{10}$$

where the Control UNets contain two standard UNets with identical architecture (Zhang et al., 2023b; Xie et al., 2023). Taking path generation as an example, the first UNet takes $(\mathbf{P}, \sigma)$ as input to perform unconditional generation, and the second takes $(\mathbf{Z}, \sigma)$ as inputs to inject goal conditions densely into the network blocks of the first one. Compared to concatenating the conditioning signals to the noise latents, this encourages close alignment between the input control and the generation. The path and full body generation models are trained in the same way as the goal model (Eq. 8), while replacing $\mathbf{Z}$ with $\mathbf{P}$ and $\mathbf{G}$. At test time, we use DDPM sampling (Ho et al., 2020) that randomly samples a Gaussian noise in the state space and iteratively denoise to the final generation.

**Ego-Perception of the World.** To generate plausible interactive behaviors, we encode the world *egocentrically* perceived by the agent, and use it to condition the behavior generation. The ego-perception code $\omega$ contains a scene code $\omega_s$, an observer code $\omega_o$, and a past code $\omega_p$, as detailed later. The ego-perception code is concatenated to the noise level $\sigma$ and passed to the denoising networks. Transforming the world to the egocentric coordinates avoids over-fitting to specific locations of the scene (Tab. 4-b), as observed in EgoPoser (Jiang et al., 2024). This also allows the model to generate novel scenarios that were not present in the training dataset. For example, there's only one data point where the cat jumps off the dining table, our method can generate diverse motion of cat jumping off the table while landing at different locations (to the left, middle, and right of the table). Please see Fig. 15 in the appendix for the corresponding visual.

**Scene, Observer, and Past Encoding.** We approximate the agent's ego-perception of the scene as its surrounding feature volume. The feature volume is queried from the 3D feature field $\Psi_s$ with Eq. 1 by transforming the sampled ego-coordinates $\mathbf{X}^a$ using the agent-to-world transformation at time $t$,

$$\omega_s = \mathrm{ResNet3D}_{\theta_\psi}(\Psi_s(\mathbf{X}^w)), \quad \mathbf{X}^w = (\mathbf{G}_t^{0,w})\mathbf{X}^a. \tag{11}$$

where $\mathrm{ResNet3D}_{\theta_\phi}$ is a 3D ConvNet with residual connections and $\omega_s \in \mathbb{R}^{64}$.

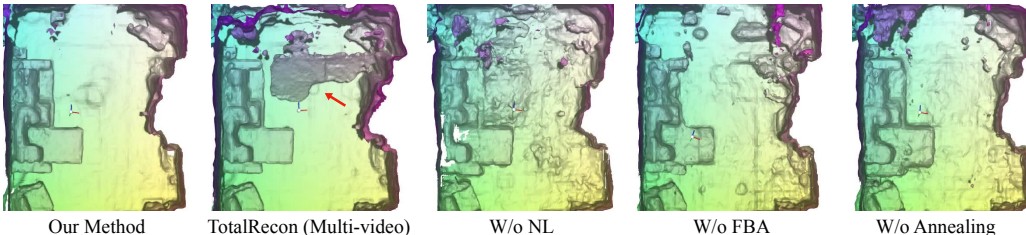

| Our Method | TotalRecon (Multi-video) | W/o NL | W/o FBA | W/o Annealing |

Figure 4: **Comparison on multi-video scene reconstruction**. We show birds-eye-view rendering of the reconstructed scene using the bunny dataset. Compared to TotalRecon that does not register multiple videos, ATS produces higher-quality scene reconstruction. Neural localizer (NL) and featuremetric losses (FBA) are shown important for camera registration. Scene annealing is important for reconstructing a complete scene from partial video captures.

Table 1: **Evaluation of Camera Registration**.

Table 2: **Dataset used in ATS**.

| Method | Rotation Error (°) | Translation Error (m) | | Videos | Length | Unique Days / Span |
|---|---|---|---|---|---|---|
| Ours | **6.35** | **0.41** | Cat | 23 | 25m 39s | 9 / 37 days |
| w/o Neural Localizer | 37.59 | 0.83 | Human | 5 | 9m 27s | 2 / 4 days |
| w/o Featuremetric BA | 22.47 | 1.30 | Dog | 3 | 7m 13s | 1 / 1 day |
| Multi-video TotalRecon | 59.19 | 0.68 | Bunny | 2 | 1m 48s | 1 / 1 day |

To encode the observer perceived by the agent, we transform the observer's past trajectory to the ego-coordinate of the agent and pass it to an MLP. We use the past $N' = 8$ frames, which corresponds to $T' = 0.8s$ from current time $t$,

$$\omega_o = \text{MLP}_{\theta_o}\left(\{\boldsymbol{\xi}_{t_i}^a\}_{i=0}^{N'-1}\right), \quad \boldsymbol{\xi}_{t_i}^a = (\mathbf{G}_t^{0,w})^{-1}\boldsymbol{\xi}_{t_i}^w, \quad t_i = t - T' + i\Delta t \tag{12}$$

where $\Delta t = 0.1s$ and $\omega_o \in \mathbb{R}^{64}$. Accounting for the external factors from the "world" enables interactive behavior generation, where the motion of an agent follows the environment constraints and is influenced by the trajectory of the observer, as shown in Fig. 5.

Similarly, we encode the root and body motion of the agent in the past $T'$ seconds,

$$\omega_p = \text{MLP}_{\theta_p}(\{\mathbf{G}_{t_i}^{\{0,\dots,B\},a}\}_{i=0}^{N'-1}), \quad \mathbf{G}_{t_i}^{\{0,\dots,B\},a} = (\mathbf{G}_t^{0,w})^{-1}\mathbf{G}_{t_i}^{\{0,\dots,B\},w}. \tag{13}$$

By conditioning on the past motion, we can generate long sequences by chaining individual ones.

## 4 EXPERIMENTS

**Dataset.** We collect a dataset that emphasizes interactions of an agent with the environment and the observer. As shown in Tab. 2, it contains RGBD iPhone video collections of 4 agents in 3 different scenes, where human and cat share the same scene. The dataset is curated to contain diverse motion of agents, including walking, lying down, eating, as well as diverse interaction patterns with the environment, including following the camera, sitting on a coach, etc.

### 4.1 4D RECONSTRUCTION OF AGENT & ENVIRONMENT

**Implementation Details.** We take a video collection of the same agent as input, and build a 4D reconstruction of the agent, the scene, and the observer. We extract frames at 10 FPS and compute augmented image measurements, including object segmentation (Yang et al., 2023b), optical flow (Yang & Ramanan, 2019), DINOv2 features (Oquab et al., 2023). We use AdamW to first optimize the environment with feature-metric loss for 30k iterations, and then jointly optimize the environment and agent for another 30k iterations with all losses in Eq. 6. Optimization takes roughly 24 hours. 8 A100 GPUs are used to optimize 23 videos of the cat data, and 1 A100 GPU is used in a 2-3 video setup (for dog, bunny, and human).

**Results of Camera Registration.** We evaluate camera registration using GT cameras estimated from annotated 2D correspondences. A visual of the annotated correspondence and 3D alignment can be

Table 3: **Evaluation of 4D Reconstruction**. SV: Single-video. MV: Multi-video.

| Method | DepthAcc (all) | DepthAcc (fg) | DepthAcc (bg) | LPIPS (all) | LPIPS (fg) | LPIPS (bg) |
|---|---|---|---|---|---|---|
| Ours | **0.708** | **0.695** | **0.703** | **0.613** | **0.609** | **0.613** |
| SV TotalRecon | 0.533 | 0.685 | 0.518 | 0.641 | 0.619 | 0.641 |
| MV TotalRecon | 0.099 | 0.647 | 0.053 | 0.634 | 0.666 | 0.633 |

found in Fig. 16 of the appendix. We report camera translation and rotation errors in Tab. 1. We observe that removing neural localization (Eq. 4) produces significantly larger localization error (*e.g.*, Rotation error: 6.35 vs 37.56). Removing feature-metric bundle adjustment (Eq. 6) also increases the error (*e.g.*, Rotation error: 6.35 vs 22.47). Our method outperforms multi-video TotalRecon by a large margin due to the above innovations.

A visual comparison on scene registration is shown in Fig. 4. Without the ability to register multiple videos, TotalRecon produces protruded and misaligned structures (as pointed by the red arrow). In contrast, our method reconstructs a single coherent scene. With featuremetric alignment (FBA) alone but without a good camera initialization from neural localization (NL), our method produces inaccurate reconstruction due to inaccurate global alignment in cameras poses. Removing FBA while keeping NL, the method fails to accurately localize the cameras and produces noisy scene structures. Finally, removing scene annealing procures lower quality reconstruction due to the partial capture.

**Results of 4D Reconstruction.** We evaluate the accuracy of 4D reconstruction using synchronized videos captured with two moving iPhone cameras looking from opposite views. The results can be found in Tab. 3. We compute the GT relative camera pose between the two cameras from 2D correspondence annotations. One of the synchronized videos is used for 4D reconstruction, and the other one is used as held-out test data. For evaluation, we render novel views from the held-out cameras and compute novel view depth accuracy DepthAcc (depth accuracy thresholded at 0.1m) for all pixels, agent, and scene, following TotalRecon. Our method outperforms both the multi-video and single-video versions of TotalRecon in terms of depth accuracy and LPIPS, due to the ability of leveraging multiple videos. A visual comparison can be found in Fig. 11. More qualitative results can be found in Fig. 6, Fig. 10 of the appendix and the supplementary webpage.

## 4.2 INTERACTIVE AGENT BEHAVIOR PREDICTION

**Data.** We train the behavior models per agent video collection. The ground-truth agent motion is extracted from the 4D reconstruction from Sec. 4.1, where we take the world-space root trajectories and bone trajectories and divide them into sequences of $T = 6.4$s. The first $T' = 0.8$s are used as input to predict the goal, path, and body motion in future $T^* = 5.6$s. The ground-truth goal is set as the position of the agent's root $T^*$s into the future, and ground-truth path and body motion are set as the trajectory of the agent's root and bones $T^*$s into the future. We use the cat dataset for quantitative evaluation, where the data are split into a training set of 22 videos and a test set of 1 video.

**Implementation Details.** Our model consists of three diffusion models, for goal, path, and full body motion respectively. To train the behavior model, we slice the reconstructed trajectory in the training set into overlapping window of $6.4$s, resulting in 12k data samples. We use AdamW to optimize the parameters of the scores functions $\{\theta_{\mathbf{Z}}, \theta_{\mathbf{P}}, \theta_{\mathbf{G}}\}$ and the ego-perception encoders $\{\theta_\psi, \theta_o, \theta_p\}$ for 120k steps with batch size 1024. Training takes 10 hours on a single A100 GPU. Each diffusion model is trained with random dropout of the conditioning (Ho & Salimans, 2022).

**Metrics.** The behavior of an agent can be evaluated along multiple axes, and we focus on goal, path, and body motion prediction. For goal prediction, we use minimum displacement error (minDE) (Chai et al., 2019). The evaluation asks the model to produce $K = 16$ hypotheses, and minDE finds the one closest to the ground-truth to compute the distance. For path and body motion prediction, we use minimum average displacement error (minADE), which are similar to goal prediction, but additionally averages the distance over path and joint angles before taking the min.

**Comparisons and Ablations.** We compare to related methods in our setup and the quantitative results are shown in Tab. 4. To predict the goal of an agent, classic methods build statistical models of how likely an agent visits a spatial location of the scene, referred to as location prior (Ziebart et al., 2009; Kitani et al., 2012). Given the extracted 3D trajectories of an agent in the egocentric coordinate, we build a 3D preference map over 3D locations as a histogram, which can be turned

Table 4: **End-to-end Evaluation of Interactive Behavior Prediction.** We report results of predicting goal, path, orientation, and joint angles, using $K = 16$ samples across $L = 12$ trials. The metrics are minADE with standard deviations ($\pm\sigma$). The lower the better and the best results are in bold.

| Method | Goal (m) $\downarrow$ | Path (m) $\downarrow$ | Orientation (rad) $\downarrow$ | Joint Angles (rad) $\downarrow$ |
|---|---|---|---|---|
| Location prior (Ziebart et al., 2009) | $0.663^{\pm0.307}$ | N.A. | N.A. | N.A. |
| Gaussian (Kendall & Gal, 2017) | $0.942^{\pm0.081}$ | $0.440^{\pm0.002}$ | $1.099^{\pm0.003}$ | $0.295^{\pm0.001}$ |
| ATS (Ours) | $\mathbf{0.448}^{\pm0.146}$ | $\mathbf{0.234}^{\pm0.054}$ | $\mathbf{0.550}^{\pm0.112}$ | $\mathbf{0.237}^{\pm0.006}$ |
| (a) hier$\rightarrow$1-stage (Tevet et al., 2022) | $1.322^{\pm0.071}$ | $0.575^{\pm0.026}$ | $0.879^{\pm0.041}$ | $0.263^{\pm0.007}$ |
| (b) ego$\rightarrow$world (Rhinehart & Kitani, 2016) | $1.164^{\pm0.043}$ | $0.577^{\pm0.022}$ | $0.873^{\pm0.027}$ | $0.295^{\pm0.006}$ |
| (c) w/o observer $\omega_o$ | $0.647^{\pm0.148}$ | $0.327^{\pm0.076}$ | $0.620^{\pm0.092}$ | $0.240^{\pm0.006}$ |
| (d) w/o scene $\omega_s$ | $0.784^{\pm0.126}$ | $0.340^{\pm0.051}$ | $0.678^{\pm0.081}$ | $0.243^{\pm0.007}$ |
| T*=4.0s (-1.6s) | $0.292^{\pm0.090}$ | $0.153^{\pm0.030}$ | $0.474^{\pm0.104}$ | $0.242^{\pm0.006}$ |
| T*=7.2s (+1.6s) | $0.579^{\pm0.122}$ | $0.330^{\pm0.048}$ | $0.539^{\pm0.061}$ | $0.246^{\pm0.006}$ |

Table 5: **Evaluation of Spatial Control.** We evaluate goal-conditioned path generation and path-conditoned full body motion generation respectively.

| Method | Path (m) $\downarrow$ | Orientation (rad) $\downarrow$ | Joint Angles (rad) $\downarrow$ |
|---|---|---|---|
| Gaussian (Kendall & Gal, 2017) | $0.206^{\pm0.002}$ | $0.370^{\pm0.003}$ | $0.232^{\pm0.001}$ |
| ATS (Ours) | $\mathbf{0.115}^{\pm0.006}$ | $\mathbf{0.331}^{\pm0.004}$ | $\mathbf{0.213}^{\pm0.001}$ |
| (a) ego$\rightarrow$world (Rhinehart & Kitani, 2016) | $0.209^{\pm0.002}$ | $0.429^{\pm0.006}$ | $0.250^{\pm0.002}$ |
| (b) control-unet$\rightarrow$code | $0.146^{\pm0.005}$ | $0.351^{\pm0.004}$ | $0.220^{\pm0.001}$ |

Figure 5: **Analysis of conditioning signals.** Removing observer and past conditioning makes the sampled goals more spread out (*e.g.*, regions both in front of and behind the agent); removing the environment conditioning introduces infeasible goals that penetrate the ground and the walls.

into probabilities and used to sample goals. Since it does not take into account of the scene and the observer, it fails to accurately predict the goal. We implement a "Gaussian" baseline that represents the goal, path, and full body motion as Gaussians, by predicting both the mean and variance of Gaussian distributions (Kendall & Gal, 2017). It is trained on the same data and takes the same input as ATS. As a result, the "Gaussian" baseline performs worse than ATS since Gaussian cannot represent multi-modal distributions of agent behaviors, resulting in mode averaging. We implement a 1-stage model similar to MDM (Tevet et al., 2022) that directly denoises body motion without predicting goals and paths (Tab. 4-a). Our hierarchical model out-performs 1-stage by a large margin. We posit hierarchical model makes it easier to learn individual modules. Finally, learning behavior in the world coordinates (Tab. 4-b) performs worse due to the over-fits to specific scene locations.

**Analyzing Interactions.** We analyse the agent's interactions with the environment and the observer by removing the conditioning signals and study their influence on behavior prediction. In Fig. 5, we show that by gradually removing conditional signals, the generated goal samples become more spread out. In Tab. 4, we drop one of the conditioning signals at a time, and find that dropping either the observer conditioning or the environment conditioning increases behavior prediction errors. We evaluated the performance with different prediction horizon $T^* = \{4.0, 5.6, 7.2\}$s and found the longer the horizon, the more difficult it is to predict the goals and future paths.

**Spatial Control.** Besides generating behaviors conditioned on agent's perception, we could also condition on user-provided spatial signals (*e.g.*, goal and path) to steer the generated behavior. The results are reported in Tab. 5. When evaluating path generation and body motion generation, the output is conditioned on the ground-truth goal and path respectively, as the goal and path $T^* = 5.6$s into the future in the 4D reconstruction. We found ATS performs better than "Gaussians" for behavior control due to its ability to represent complex distributions. Furthermore, egocentric representation produces better behavior generation results. Finally, replacing control-unet architecture by concatenating spatial control with perception codes produces worse alignment (*e.g.*, Path error: 0.115 vs 0.146).

## 5 CONCLUSION

We have presented a method for learning interactive behavior of agents grounded in 3D environments. Given multiple casually-captured video recordings, we build persistent 4D reconstructions including the agent, the environment, and the observer. Such data collected over a long time period allows us to learn a behavior model of the agent that is reactive to the observer and respects the environment constraints. We validate our design choices on casual video collections, and show better results than prior work for 4D reconstruction and interactive behavior prediction.

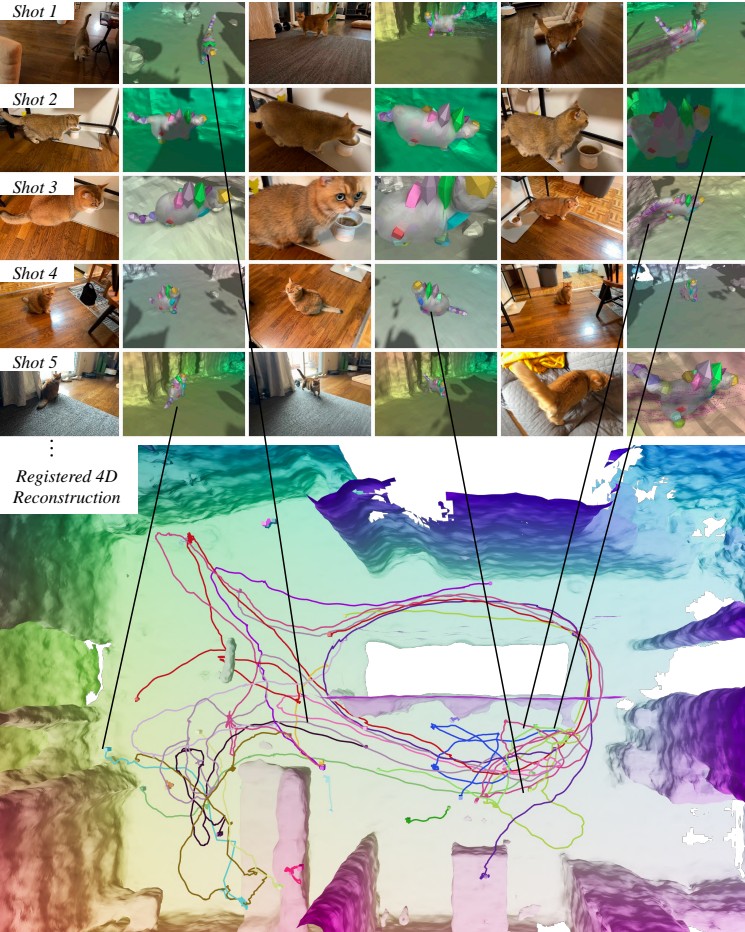

Figure 6: **Results of 4D reconstruction**. Top: reference images and renderings. Background color represents correspondence. Colored blobs on the cat represent $B = 25$ bones (*e.g.*, head is represented by the yellow blob). The magenta colored lines represents reconstructed trajectories of each blob in the world space. Bottom: Bird's eye view of the reconstructed scene and agent trajectories, registered to the same scene coordinate. Each colored line represents a unique video sequence where boxes and spheres indicate the starting and the end location.

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

# A  APPENDIX

## A.1  COMPOSITE VOLUME RENDERING

To render an image, we use composite volume rendering (Niemeyer & Geiger, 2021), where we sample rays in the camera space at time $t$, use $\mathcal{D}$ to map them to the canonical spaces of the scene and the agent, and query values (*e.g.*, density $\rho$, color $\mathbf{c}$, feature $\psi$) from the canonical fields. Their values are composed in ray integration. The composite density $\rho_i$ at sample $i$ along the ray is computed as the sum of the scene and agent's density, and the composite color $\mathbf{c}_i$ is computed as the weighted sum of each component's color,

$$\rho_i = \rho_s + \rho_a, \quad \mathbf{c}_i = \frac{\rho_s \mathbf{c}_s + \rho_a \mathbf{c}_a}{\rho_i}. \tag{14}$$

We can then use volume rendering equations (Mildenhall et al., 2020) to compute a color image,

$$\mathbf{c} = \sum_{i=1}^{N} \tau_i \alpha_i \mathbf{c}_i, \quad \tau_i = \prod_{k=1}^{i-1} (1 - \alpha_k), \quad \alpha_i = 1 - e^{-\rho_i \delta_i}, \tag{15}$$

where $N = 128$ is the number of sampled points along camera ray, $\tau_i$ is the transmittance, $\alpha_i$ is the alpha value for sample point $i$ and $\delta_i$ is the distance between sample point $i$ and the $(i+1)$. The same method can be applied to render feature images by replacing color values with features.

## A.2  DETAILS ON MODEL AND DATA

**Illustration figures.** Fig. 7 shows the representation of the agents. Fig. 8 illustrates our coarse-to-fine multi-video registration method. We provide Fig. 9 to illustrate the training of the neural scene localizer as well as the agent pose regressor.

**Table of Notation.** A table of notation used in the paper can be found in Tab. 6.

**Summary of I/O.** A summary of inputs and outputs of the method is shown in Tab. 7

**Data Collection.** We collect RGBD videos using an iPhone, similar to TotalRecon (Song et al., 2023). To train the neural localizer, we use Polycam to take the walkthrough video and extract a textured mesh. For behavior capture, we use Record3D App to record videos and extract color images and depth images.

**Data processing.** We extract frames from the videos at 10 FPS, and use off-the-shelf models to produce augmented image measurements, including optical flow (Yang & Ramanan, 2019) and DINOv2 features (Oquab et al., 2023). To get object segmentation, we use Grounding DINO (Liu et al., 2023) to annotate a bounding box given text description of the agent (e.g., cat), and SAM (Kirillov et al., 2023) to segment the agent in the first frame of the video. The segmentation is tracked over all the frames using XMem (Cheng & Schwing, 2022; Yang et al., 2023b). The pre-processing code is taken from an open-source project (Yang et al., 2023a).

**Diffusion Model Architecture.** The score function of the goal is implemented as 6-layer MLP with hidden size 128. The the score functions of the paths and body motions are implemented as 1D UNets taken from GMD (Karunratanakul et al., 2023). The sampling frequency is set to be 0.1s, resulting a sequence length of 56. The environment encoder is implemented as a 6-layer 3D ConvNet with kernel size 3 and channel dimension 128. The local feature volume is queried with a grid $\mathbf{X}^a \in \mathbb{R}^{64 \times 8 \times 64}$, which encodes a $6.4\,\mathrm{m} \times 0.8\,\mathrm{m} \times 6.4\,\mathrm{m}$ box around the agent along the width (X), height (Y), and length (Z) dimension. The observer encoder and history encoder are implemented as a 3-layer MLP with hidden size 128.

**Diffusion Model Training and Testing.** We use a linear noise schedule at training time and 50 denoising steps. We train all the diffusion models (goal, path and pose) with classifier-free guidance (Ho & Salimans, 2022; Tevet et al., 2022) that randomly sets conditioning signals to zeros $\mathbf{Z} = \varnothing$ randomly. This allows us to control the trade-off between interactive behavior and unconditional behavior generation, as shown in Fig. 14. At test time, each goal denoising step takes 2ms and each path/body denoising step takes 9ms on an A100 GPU.

**Camera Pose Annotations for Evaluation.** We annotate GT camera poses from 2D correspondence annotations. The relative camera pose is computed as follows: 1) We manually annotate seven pairs

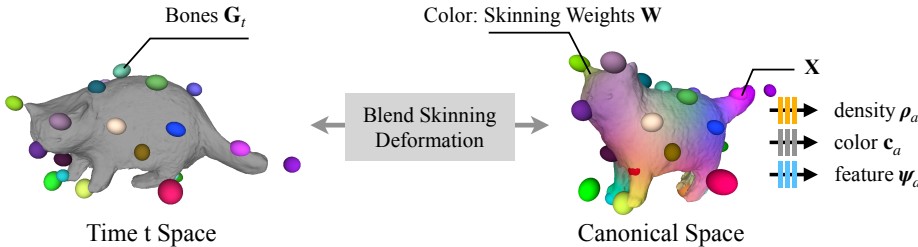

Figure 7: Agent Representation. We use BANMo (Yang et al., 2022) to obtain the deformation model of the agent, which optimizes for the canonical NeRF, articulated bones $\mathbf{G}_t$, as well as as well as the skinning $\mathbf{W}$ weights using inverse rendering.

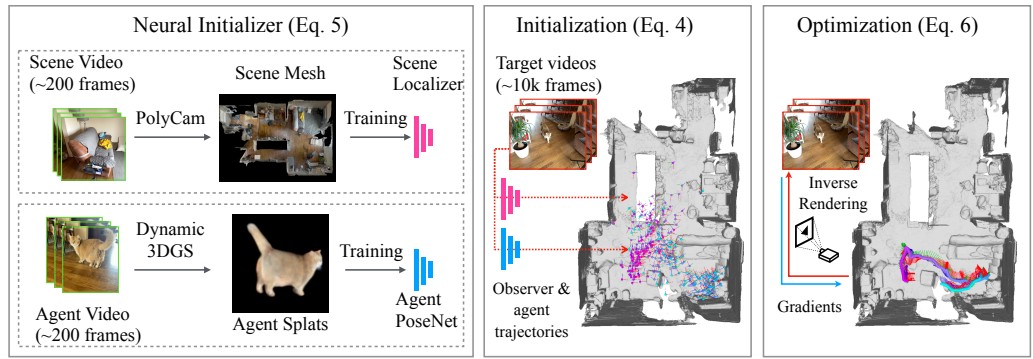

Figure 8: Coarse-to-fine Registration. 1) Given template recordings, including a walk-through video of the scene and a video that covers enough views of the agent, we build template reconstruction of the environment and the agent using existing algorithms, Polycam and Lab4d (Yang et al., 2023a). 2) Then we use the 3D reconstructions as data generator to train a scene localizer and an agent pose prediction network. 3) Given a collection of target videos to reconstruct, we use the neural localizers to initialize the corresponding scene and agent camera poses, and jointly optimized them with the canonical neural fields and motion parameters.

of 2D point correspondences between the two frames; 2) 2D points are then back-projected to 3D give the depth map from iPhone; 3) We solve Procrustes registration between two sets of corresponding 3D points to obtain relative camera poses. A visual of the annotated correspondence and 3D alignment can be found in Fig. 16 of the appendix.

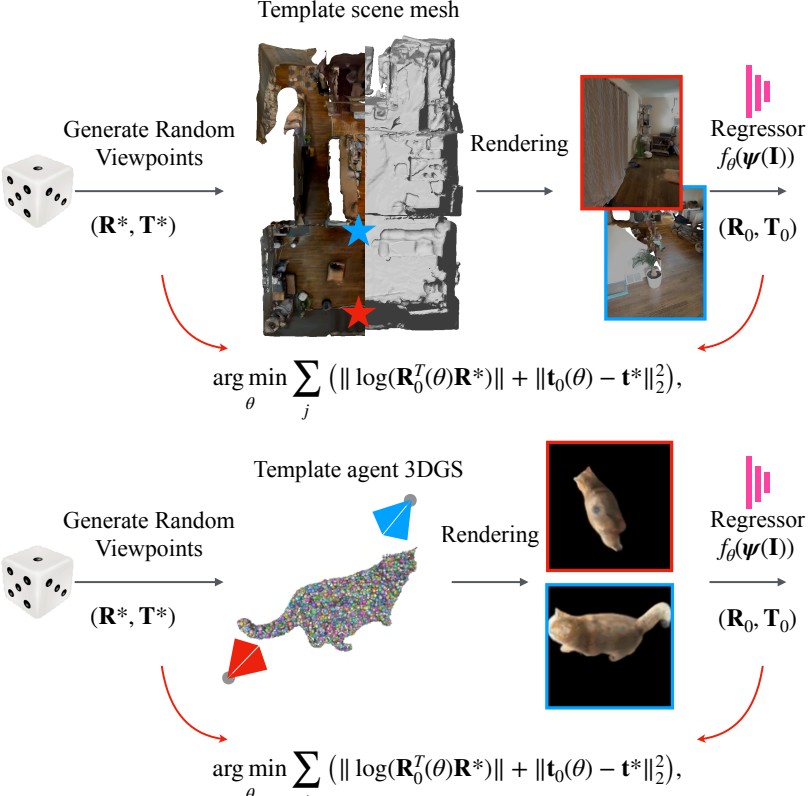

Figure 9: Training neural scene localizer (top) and agent pose regressor (bottom). Given a template 3D mesh of the scene or tempalte Gaussians splats (3DGS) of an agent, we generate random camera viewpoints and render images on the fly. The networks are trained to regress the camera rotation and translation given DINO-v2 features of a single image $\psi(\mathbf{I})$. The model is trained for 8k iterations with batchsize of 128. This mechanism enables us to obtain good initialization for the input videos to register them in the consistent world coordinate frame.

Table 6: Table of Notation.

| Symbol | Description |
|--------|-------------|
| **Global Notations** | |
| $B$ | The number of bones of an agent. By defatult $B = 25$. |
| $M$ | The number of videos. |
| $N_i$ | The number of image frames extracted from video $i$. |
| $\mathbf{I}_i$ | The sequence of color images $\{I_1, \ldots, I_{N_i}\}$ extracted from video $i$. |
| $\boldsymbol{\psi}_i$ | The sequence of DINOv2 feature images $\{\boldsymbol{\psi}_1, \ldots, \boldsymbol{\psi}_{N_i}\}$ extracted from video $i$. |
| $T_i$ | The length of video $i$. |
| $T^*$ | The time horizon of behavior diffusion. By default $T^* = 5.6$s. |
| $T'$ | The time horizon of past conditioning. By default $T' = 0.8$s |
| $\mathbf{Z} \in \mathbb{R}^3$ | Goal of the agent, defined as the location at the end of $T^*$. |
| $\mathbf{P} \in \mathbb{R}^{3 \times T^*}$ | Path of the agent, defined as the root body trajectory over $T^*$. |
| $\mathbf{G} \in \mathbb{R}^{6B \times T^*}$ | Pose of the agent, defined as the 6DoF rigid motion of bones over $T^*$. |
| $\omega_s \in \mathbb{R}^{64}$ | Scene code, representing the scene perceived by the agent. |
| $\omega_o \in \mathbb{R}^{64}$ | Observer code, representing the observer perceived by the agent. |
| $\omega_p \in \mathbb{R}^{64}$ | Past code, representing the history of events happened to the agent. |
| **Learnable Parameters of 4D Reconstruction** | |
| $\mathbf{T}$ | Canonical NeRFs, including a scene MLP and an agent MLP. |
| $\boldsymbol{\beta}_i \in \mathbb{R}^{128}$ | Per-video code that allows NeRFs to represent variations across videos. |
| $\mathcal{D}$ | Time-varying parameters, including $\{\boldsymbol{\xi}, \mathbf{G}, \mathbf{W}\}$. |
| $\boldsymbol{\xi}_t \in SE(3)$ | The camera pose that transforms the scene to the camera coordinates at t. |
| $\mathbf{G}_t^0 \in SE(3)$ | The camera pose that transforms the canonical agent to the camera coordinates at t. |
| $\mathbf{G}_t^b \in SE(3)$ | The transformation that moves bone b from its rest state to time t state. |
| $\mathbf{W} \in \mathbb{R}^B$ | Skinning weights of a point, defined as the probability of belonging to bones. |
| $f_\theta$ | PoseNet that takes a DINOv2 feature image as input and produces camera pose. |
| **Learnable Parameters of Behavior Generation** | |
| $\mathrm{MLP}_{\theta_\mathbf{Z}}$ | Goal MLP that represent the score function of goal distributions. |
| $\mathrm{ControlUNet}_{\theta_\mathbf{P}}$ | Path UNet that represents the score function of path distributions. |
| $\mathrm{ControlUNet}_{\theta_\mathbf{G}}$ | Pose UNet that represents the score function of pose distributions. |
| $\mathrm{ResNet3D}_{\theta_\psi}$ | Scene perception network that produces $\omega_s$ from 3D feature grids $\boldsymbol{\psi}$. |
| $\mathrm{MLP}_{\theta_\mathbf{o}}$ | Observer MLP that produces $\omega_o$ from observer's past trajectory in $T'$. |
| $\mathrm{MLP}_{\theta_\mathbf{P}}$ | Past MLP that produces $\omega_p$ from agent's past trajectory in $T'$. |

Table 7: Summary of inputs and outputs at different stages of the method.

| Stage | Description |
|-------|-------------|
| Overall | Input: A walk-through video of the scene and videos with agent interactions. 
 Output: An interactive behavior generator of the agent. |
| Localizer Training | Input: 3D reconstruction of the environment and the agent. 
 Output: Neural localizer $f_\theta$. |
| Neural Localization | Input: Neural localizer $f_\theta$ and the agent interaction videos. 
 Output: Camera poses for each video frame. |
| 4D Reconstruction | Input: A collection of videos and their corresponding camera poses. 
 Output: Scene feature volume $\boldsymbol{\Psi}$, motion of the agent $\mathbf{G}$ and observer $\boldsymbol{\xi}$. |
| Behavior Learning | Input: Scene feature volume $\boldsymbol{\Psi}$, motion of the agent $\mathbf{G}$ and observer $\boldsymbol{\xi}$. 
 Output: An interactive behavior generator of the agent. |

### A.3 Additional Results

**Results on more agents.** In the main paper, we show results for cat videos. We show results on more agents, including a human, a dog, and a bunny in the supplementary webpage.

**Comparison to TotalRecon.** In the main paper, we compare to TotalRecon on scene reconstruction by providing it multiple videos. Here, we include additional comparison in their the original single video setup. We find that TotalRecon fails to build a good agent model, or a complete scene model given limited observations, while our method can leverage multiple videos as inputs to build a better agent and scene model. The results are shown in Fig. 10.

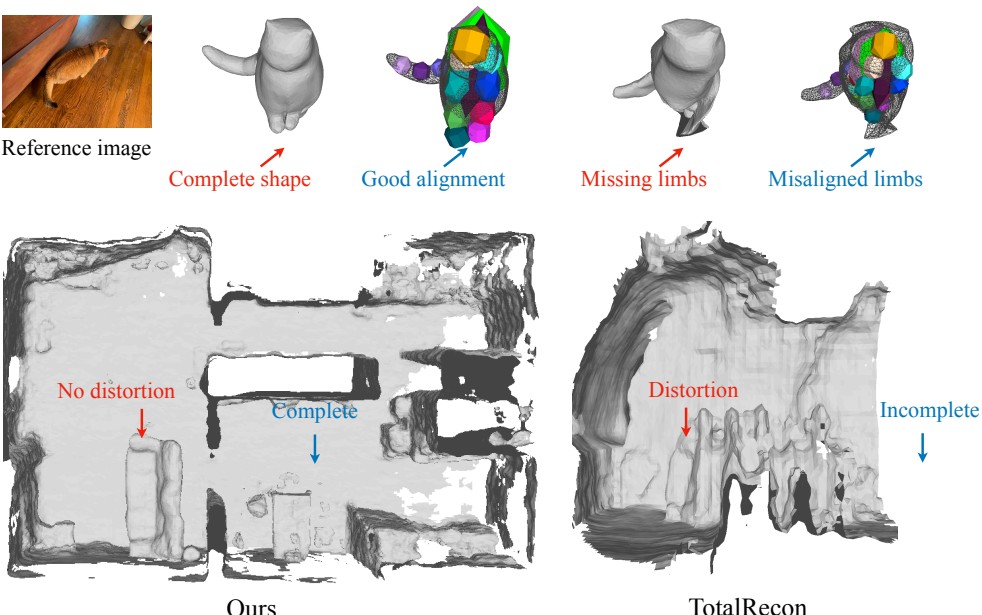

Figure 10: Qualitative comparison with TotalRecon (Song et al., 2023) on 4D reconstruction. Top: reconstruction of the agent at at specific frame. Total-recon produces shapes with missing limbs and bone transformations that are misaligned with the shape, while our method produces complete shapes and good alignment. Bottom: reconstruction of the environment. TotalRecon produces distorted and incomplete geometry (due to lack of observations from a single video), while our method produces an accurate and complete environment reconstruction.

**Visual Ablation on Scene Awareness.** We show final camera and agent registration to the canonical scene in Fig. 13. The registered 3D trajectories provides statistics of agent's and user's preference over the environment.

**Histogram of Agent / Observer Visitation.** We show final camera and agent registration to the canonical scene in Fig. 12. The registered 3D trajectories provides statistics of agent's and user's preference over the environment.

### A.4 Limitations and Future Works

**Environment Reconstruction.** To build a complete reconstruction of the environment, we register multiple videos to a shared canonical space. However, the transient structures (e.g., cushion that can be moved over time) may not be reconstructed well due to lack of observations. We notice displacement of chairs and appearance of new furniture in our capture data. Our method is robust to these in terms of camera localization (Tab. 1 and Fig. 17). However, 3D reconstruction of these transient components is challenging. As shown in Fig 17, our method fails to reconstruct notable layout changes when they are only observed in a few views, e.g., the cushion and the large boxes

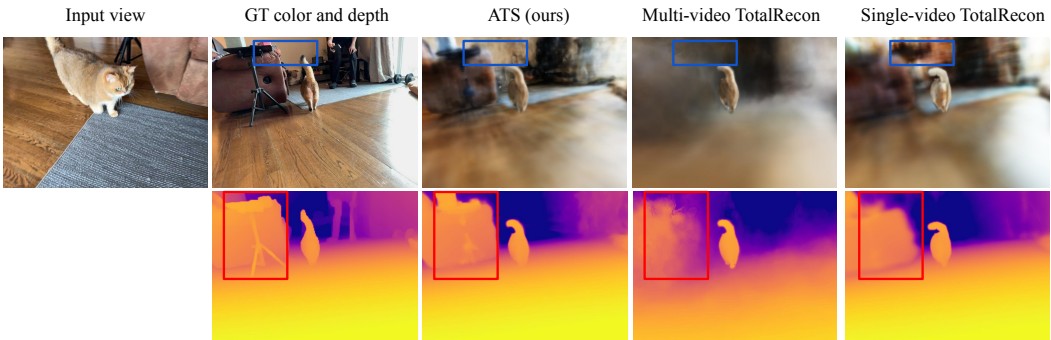

Figure 11: **Qualitative comparison on 4D reconstruction (Tab. 3).** We compare with TotalRecon on 4D reconstruction quality. We show novel views rendered with a held-out camera that looks from the opposite side. ATS is able to leverage multiple videos captured at different times to reconstruct the wall (blue box) and the tripod stand (red box) even they are not visible in the input views. Multi-video TotalRecon produces blurry RGB and depth due to bad camera registration. The original TotalRecon takes a single video as input and therefore fails to reconstruct the regions (the tripod and the wall) that are not visible in the input video.

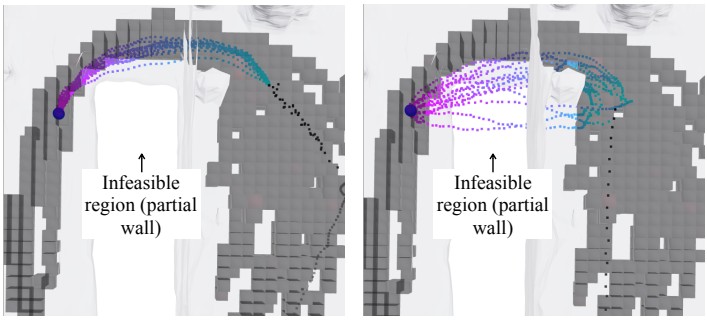

Figure 12: **Visual ablation on scene awareness.** We demonstrate the effect of the scene code $\omega_s$ through goal-conditioned path generation (bird's-eye-view, blue sphere→goal; gradient color→generated path; gray blocks→locations that have been visited in the training data). Conditioned on scene, the generated path abide by the scene geometry, while removing the scene code, the generated paths go through the wall in between two empty spaces.

(left) and the box (right). We leave this as future work. Leveraging generative image prior to in-paint the missing regions is a promising direction to tackle this problem (Wu et al., 2023).

**Scaling-up.** We demonstrate our approach on four types of agents with different morphology living in different environments. For the cat, we use 23 video clips over a span of a month. This isn't large-scale but we believe this is an important step to go beyond a single video. In terms of robustness, we showed a meaningful step towards scaling up 4D reconstruction by neural initialization (Eq. 6). The major difficulty towards large-scale deployment is the cost and robustness of 4D reconstruction using test-time optimization.

**Multi-agent Interactions.** ATS only handles interactions between the agent and the observer. Interactions with other agents in the scene are out of scope, as it requires data containing more than one agent. Solving re-identification and multi-object tracking in 4D reconstruction will enable introducing multiple agents. We leave learning multi-agent behavior from videos as future work.

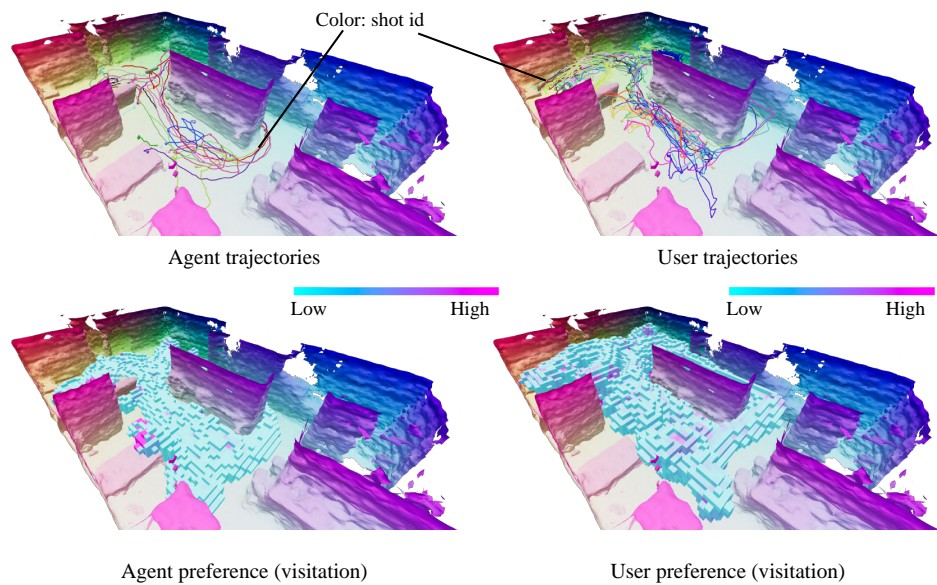

Figure 13: Given the 3D trajectories of the agent and the user accumulated over time (top), one could compute their preference represented by 3D heatmaps (bottom). Note the high agent preference over table and sofa.

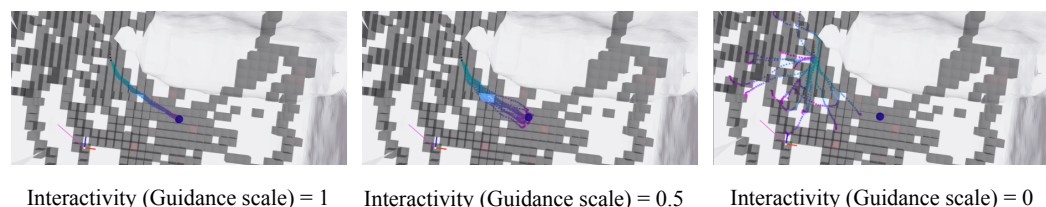

Figure 14: **Interactivity of the agent**. By changing the classifier-free guidance scale $s$, we can find a trade-off between interactive behavior and unconditional behavior. We demonstrate the control over interactivity by goal-conditioned path generation (bird's-eye-view, blue sphere→goal; gradient color→generated path). With a higher classifier-free guidance scale $s$, the model is controlled more by the conditional generator, and therefore exhibits higher interactivity. $s = 0$ corresponds to fully unconditional generation.

**Complex Scene Interactions.** Our approach treat the background as a rigid component without accounting for movable and articulated scene structures, such as doors and drawers. To reconstruct complex interactions with the environment, one approach is to extend the scene representation to be hierarchical (with a kinematic tree), such that it consists of articulated models of interactable objects. To generate plausible interactions between the agent and the scene (e.g., opening a door), one could extend the agent representation $G$ to include both the agent and the articulated objects (e.g., door).

**Physical Interactions.** Our method reconstructs and generates the kinematics of an agent, which may produce physically-implausible results (e.g., penetration with the ground and foot sliding). One promising way to deal with this problem is to add physics constraints to the reconstruction and motion generation (Yuan et al., 2023).

**Long-term Behavior.** The current ATS model is trained with time-horizon of $T^* = 6.4$ seconds. We observe that the model only learns mid-level behaviors of an agent (e.g., trying to move to a destination; staying at a location; walking around). We hope incorporating a memory module and training with longer time horizon will enable learning higher-level behaviors of an agent.

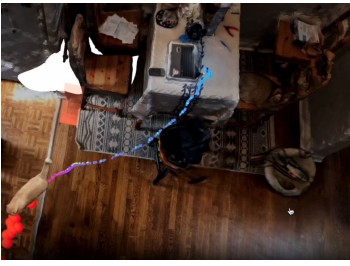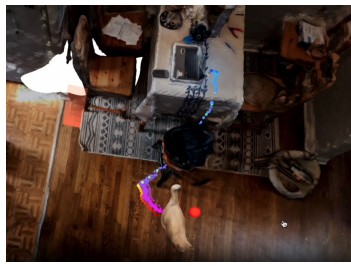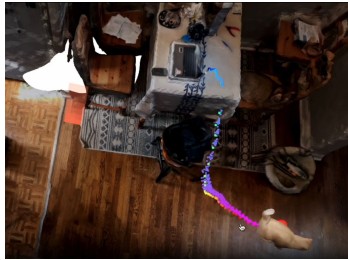

Figure 15: **Generalization ability of the behavior model.** Thanks to the ego-centric encoding design (Eq. 12), a specific behavior can be learned and generalized to novel situations even it was seen once. Although there's only one data point where the cat jumps off the dining table, our method can generate diverse motion of cat jumping off the table while landing at different locations (to the left, middle, and right of the table) as shown in the visual.

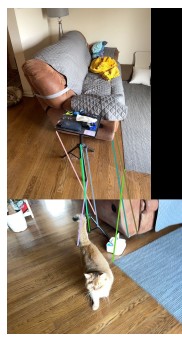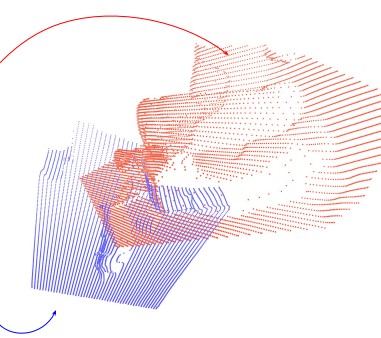

Figure 16: **GT correspondence and 3D alignment.** Left: Annotated 2D correspondence between the canonical scene (top) and the input image (bottom). Right: we visualize the GT camera registration by transforming the input frame 3D points (blue, back-projected from depth) to the canonical frame (red). The points align visually.

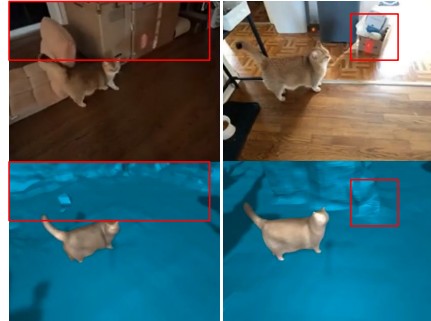

Figure 17: **Robustness to layout changes.** We find our camera localization to be robust to layout changes, e.g., the cushion and the large boxes (left) and the box (right). However, it fails to *reconstruct* layout changes, especially when they are only observed in a few views.

## A.5 SOCIAL IMPACT

Our method is able to learn interactive behavior from videos, which could help build simulators for autonomous driving, gaming, and movie applications. It is also capable of building personalized behavior models from casually collected video data, which can benefit users who do not have access to a motion capture studio. On the negative side, the behavior generation model could be used as "deepfake" and poses threats to user's privacy and social security.

