# OpenReview forum: "Agent-to-Sim: Learning Interactive Behavior Models from Casual Longitudinal Videos"
_ICLR.cc/2025/Conference — ICLR 2025 Poster_

### Official Review · Reviewer_8uBB · 2024-10-31

**Soundness:** 3
**Presentation:** 3
**Contribution:** 3
**Rating:** 8
**Confidence:** 2

**Summary:**

This paper proposes a 4D reconstruction method and environment-aware behavior generation method for 3D agents.  It represents the canonical structures for the scene and agents by MLPs as in NeRF, and represents the time-varying structure with camera pose, root pose and bone transformations at each frame. These structures  are all learned from casually captured videos. After recovering the structures, three diffusion models are trained to generate goal, path and body poses conditioned the 3D feature field of the scene.

The main contribution of this paper is a complete pipeline to reconstruct and simulate interactive behaviors of 3D agents in a dynamic scene.

**Strengths:**

I appreciate the careful 4D agent-to-sim pipeline design, including canonical and time varying structure design, the camera localization and the optimization procedure. It enables the improvement of the interactive behavior simulation results as verified in the rendering quality and Interactive behavior prediction.

1. the simulated behaviors or animations of cats and humans looks realistic and interacts with the scene geometry in a physics-plausible way, no obvious penetration between the agent and the scene geometry in the video demos.
2. The hierarchical behavior simulation model, from goal, path to body poses, and egocentric perception features in this paper are verified to be effective in improving the accuracy of the behavior prediction.

**Weaknesses:**

1.  It is not clear how to collect the training data for neural camera localization. Does the camera move along with the agent or fixed during capturing？  If it is fixed, there are only a few camera poses that can be used in the training.  If it can move, that means we already have an algorithm to obtain accurate camera poses for training.
2. The symbol \sigma is used to represent density in Sec. 3.1 and noise in Sec. 3.3, a kind of misleading when reading the paper.

**Questions:**

I have some questions  on the paper writing:

1. In eq.8, both \sigma and \epsilon indicate the noise?  since it is mentioned in line 303 that \sigma is a noise value. If it is the case, please clarify: Does \epsilon indicate the variance of the noise?
2. In line 310,  how large is the volume queried from the 3D feature volume?
3. What is the definition of the generated path? Is it the root translations in the scene？

---

> ### Author Response · Authors · 2024-11-22
>
> Thanks for the encouraging comments and constructive feedback. We have revised the paper to address the confusions about neural camera localization, notations, and method details. Key text changes are highlighted in **dark red** in the [revision](https://openreview.net/pdf?id=y80D4IojuY).
>
> Below, we provide answers to the questions raised and would welcome any further feedback or clarification requests.
>
> Q1.
> > It is not clear how to collect the training data for neural camera localization. Does the camera move along with the agent or fixed during capturing？ If it is fixed, there are only a few camera poses that can be used in the training. If it can move, that means we already have an algorithm to obtain accurate camera poses for training.
>
> The paired GT poses and images are synthetic data generated from a 3D map of the room. The 3D map is built using Polycam from a **single video, without the agent**. During training, we generate random camera viewpoints and render images on the fly given the 3D map, and the localizer learns to regress the camera rotation and translation from a DINO image. Then, neural localizers are used to initialize the optimization in Eq. 6 **across all the videos**. This mechanism enables us to obtain good initialization for the target videos to register them in the **consistent world coordinate**. We revised the paper in **L224-237** accordingly, and added **Fig. 8 and Fig.9 in the appendix** to illustrate the process of training neural localizers and using them to initialize optimization on target videos.
>
> Q2.
> > The symbol \sigma is used to represent density in Sec. 3.1 and noise in Sec. 3.3, a kind of misleading when reading the paper.
>
> Thanks for pointing it out. We have replaced $\sigma$ with $\rho$ to denote density in the revision to reduce confusion.
>
> Q3.
> > In eq.8, both \sigma and \epsilon indicate the noise? since it is mentioned in line 303 that \sigma is a noise value. If it is the case, please clarify: Does \epsilon indicate the variance of the noise?
>
> $\sigma$ is the standard deviation of the Gaussian noise (dependent on the timestep in the diffusion process) and $\epsilon$ is the actual noise that is sampled from the gaussian distribution. We have clarified in **L306** of the revision.
>
> Q4.
> > In line 310, how large is the volume queried from the 3D feature volume?
>
> The local feature volume is queried with a grid ${\bf X}^a \in \mathbb{R}^{64\times8\times64}$, which encodes a $6.4\mathrm{m} \times 0.8\mathrm{m} \times 6.4\mathrm{m}$ box around the agent along the width (X), height (Y), and length (Z) dimension. This has been included in **L854-855** of the revision.
>
> Q5.
> > What is the definition of the generated path? Is it the root translations in the scene？
>
> This is correct and it has been clarified in **L261-264** of the revision.
>
> We hope these revisions and explanations address the concerns raised. Please let us know if there are further questions or aspects needing clarification.

---

> > ### Comment · Reviewer_8uBB · 2024-11-22
> > **The rebuttal addresses my concerns**
> >
> > Thanks for the rebuttal.   I have checked the clarifications made in the revision, and they addresses my concerns in review comments. I will raise my score to acceptance.

---

### Official Review · Reviewer_C85q · 2024-11-02

**Soundness:** 2
**Presentation:** 2
**Contribution:** 3
**Rating:** 5
**Confidence:** 3

**Summary:**

This paper proposes a method for learning interactable behaviour of agent from casual video. To this end, the reconstruct the video (agent, scene and observer) in 4D. The representation has canonical structure in 3D for time independent element using NERF and a time varying structure (observer and agent). The paper proposes a coarse to fine registration of agent and environment into a canonical 4D space. This 4D representation further helps in behaviour representation further used in understanding the behaviour of agents and goal conditioned trajectory generation.

**Strengths:**

The paper shows an original pipeline to reconstruct 4D representation from multiple videos over longer timespan like a month. This is a complicated problem and the method solves this with good qualitative and quantitative results.
The choice of representation in every stages is carefully thought through and use of the body of knowledge is good. For example, the study by Banani et.al, which shows that large image models have good 3D and viewpoint awareness, is exploited nicely in this paper.
The decoupling of static and dynamic structure and registration is well designed and merged to solve the complicated relationships exist in a long scene with different variation in layout or color over time.

**Weaknesses:**

Despite the strength, the paper writing is not very good for a reader. Many places the approach is not well understood. In many places argument is placed without much conviction, for example, neural localiser finds more robust than geometric correspondence while being computationally efficient is not well grounded through technical principles or empirical evidences. The ego perception of the world and scene, observer, and past encoding needs more elaboration in writing to make them understand. It is not clear how the path is generated given a goal? is this a shortest path or any other path. More doubts are raised as questions below.

Because the writing, required illustrations, and the flow of thought is bit convoluted, I am tending towards the low score of the paper.

**Questions:**

1. I assume Equation 1 and 2 are through NERF context. it is not clear to train them, how to get the poses to train them?
2. How to train Equation 2 when agent is in motion. Elaboration is needed.
3. A picture is needed to show the representation of agent. It is not clear what is meant by root and how that is identified from video automatically for any morphology.
4. Satio et. al. uses SMPL morphology where getting the skinning is understood. Here the agent can be anything. So the skinning part needs more elaboration to understand by the reader.  What is meant by "bones have time varying centers" and how this impact the skinning? Need picture with explanation.
5. What do we mean by set of 3D Gaussian as bones? How they are fitted in video? How they are initialised?
6.  Scene specific neural localiser  regresses the camera poses, how to get the GT poses? In the following paragraph it is written that "to learn neural localiser we first capture a walkthrough video and build a 3D map.." how to build this 3D map? where the poses are coming from? Is this video is the same we are dealing for 4D reconstruction? Lots of confusion here.
7.If the above 3D is in place , we can sample R*, T* and solve Eqn 5. But what is theta there and what is the importance of that parameter? Agent can move while R* can be fixed.  This is not well understood.
8. Dynamic 3DGS also require poses. How to get them?  Equation 5 needs more illustrative explanation.
9. Because the writing of registration section is bit convoluted questions are arising for Behavious representation and Goal generation. Why score based method was needed for generating goal? VAE could be good enough?
10. Scene observer and past encoding is a complex design. When the scene is already encoded as a MLP(Eq 1,2) it is not clear why another encoding? Or am I missing something?

---

> ### Author Response · Authors · 2024-11-22
>
> Thanks for your constructive feedback. Based on the comments, we have revised the paper to address the noted weaknesses in writing. Key text changes are highlighted in **dark red** in the [revision](https://openreview.net/pdf?id=y80D4IojuY). Below, we provide detailed answers to the questions raised and would welcome any further feedback or clarification requests.
>
> Below, we answer the questions about technical details.
>
> Q1.
> > “Neural localizer is found to be more robust than geometric correspondence while being computationally efficient” is not well grounded through technical principles or empirical evidence.
>
> Geometric correspondence methods, such as DUSt3R (Wang et al., 2023), scale with $\mathcal{O}(N^2)$ memory and computation for $N$ images, which becomes infeasible for large-scale datasets (e.g., 10k images). In contrast, registering each image to a canonical representation reduces the cost to $\mathcal{O}(N)$, making it significantly more efficient and feasible to run at scale. We clarified this point in **L221–224**.
>
> Q2.
> > The ego perception of the world and scene, observer, and past encoding needs more elaboration in writing to make them understand. It is not clear how the path is generated given a goal? Is this the shortest path or any other path?
>
> The path represents the root joint locations generated by our diffusion model. At test time, we apply DDPM sampling: initializing with Gaussian noise in trajectory space and iteratively denoising it using the learned score functions (Eq. 9). The training of the path and body diffusion models follows the same procedure as the goal diffusion model. We expanded the description in **L290–302**.
>
> Q3.
> > I assume Equation 1 and 2 are through NERF context. it is not clear to train them, how to get the poses to train them?
>
> Our 4D representation (scene NeRFs and agent NeRFs, and deformation fields) is optimized using **Eq. 6**. The scene camera poses are initialized from the neural localizers and the agent camera poses are initialized from the pose regressor. The optimization leverages differentiable rendering to optimize NeRFs and deformation fields against video observations, including color, flow, and depth. We revised **L195-201** and added further details on compositional rendering in **Appendix A.1**. We also added **Fig. 8 in the appendix** to better illustrate the process to obtain and use neural localizers to initialize optimization on target videos.
>
> Q4.
> > How to train Equation 2 when agent is in motion. Elaboration is needed. A picture is needed to show the representation of agent. It is not clear what is meant by root and how that is identified from video automatically for any morphology.
>
> We use BANMo to learn the deformation model of the agent, as clarified in **L181–194**. BANMo jointly optimizes bone articulations and skinning parameters, where the "root" is the canonical agent coordinate, typically the agent's center.  An illustration figure of the agent representation is included in the **Appendix Fig. 7**.
>
> Q5.
> > Saito et. al. uses SMPL morphology where getting the skinning is understood. Here the agent can be anything. So the skinning part needs more elaboration to understand by the reader. What is meant by "bones have time varying centers" and how this impact the skinning? Need picture with explanation. What do we mean by set of 3D Gaussian as bones? How they are fitted in video? How they are initialized?
>
> We use BANMo's skinning model. Given the bone locations and scales, skinning weights are computed as the Mahalanobis distances between a point and bones, normalized by Softmax. The bones are initialized uniformly on a sphere and optimized with inverse rendering. We revised the paper in **L181–194** and a visual illustration can be found in **Appendix Fig. 7**.

---

> > ### Author Response · Authors · 2024-11-22
> >
> > Q6.
> > > Scene specific neural localiser regresses the camera poses, how to get the GT poses? In the following paragraph it is written that "to learn neural localiser we first capture a walkthrough video and build a 3D map.." how to build this 3D map? where the poses are coming from? Is this video is the same we are dealing for 4D reconstruction? Lots of confusion here. 7.If the above 3D is in place , we can sample R*, T* and solve Eqn 5. But what is theta there and what is the importance of that parameter? Agent can move while R* can be fixed. This is not well understood.
> >
> > The paired GT poses and images are synthetic data generated from a 3D map of the room. The 3D map is built using Polycam from a **single video**, **without the agent**. This was specified in L855 appendix of the original submission, and we pulled this earlier to L226 in the revision. The neural localizer is parameterized with $\theta$ and it is a ResNet-18 with a head that regresses (R, T). The neural localizers are used to initialize the optimization in **Eq. 6** across **all the videos**. We revised the paper in **L224-237** accordingly, and added **Fig. 8 and Fig.9 in the appendix** to better illustrate the process to train neural localizers and use them to initialize optimization on target videos.
> >
> > Q7.
> > > Dynamic 3DGS also require poses. How to get them? Equation 5 needs more illustrative explanation.
> >
> > Thanks for asking. Essentially we follow BANMo, which fits NeRF and deformation fields to a single video. Please note that we refer to this stage as Dynamic 3DGS but we're not referring to the paper which has the same name (Luiten et al., 2024). We use Lab4D (Yang et al., 2023a), which implements BANMo with dynamics 3DGS to post-process the NeRF into gaussian splats for fast rendering.
> >
> > Q8.
> > > Because the writing of the registration section is a bit convoluted, questions are arising for Behavior representation and Goal generation. Why was the score based method needed for generating a goal? Could VAE be good enough?
> >
> > VAE is a valid alternative; however, we opted for diffusion models, which is used in most recent human motion generation works (e.g., [Tevet et al., 2022]). We found it to be easy to train, use, and expressive in its ability to capture multimodal distributions.
> >
> > Q9.
> > > Scene observer and past encoding is a complex design. When the scene is already encoded as a MLP(Eq 1,2) it is not clear why another encoding? Or am I missing something?
> >
> > While the scene is encoded by Eq 1., we still need a representation that specifies where the agent is, relative to the scene, for the behavior prediction network. Hence we encode it using an ego-centric coordinate frame, which captures the local surrounding of the agent to predict its future behavior. Similarly, the representation of the observer relative to the scene has to be converted into the input for the prediction network.
> >
> > We hope these revisions and explanations address the concerns raised. Please let us know if there are further questions or aspects needing clarification.

---

> ### Author Response · Authors · 2024-11-22
>
> .

---

> ### Comment · Reviewer_C85q · 2024-11-23
> **Thanks for the clarification**
>
> Thanks for the clarifications.
>
> If we know the M videos are all corresponds to the 3D map generated using Polycam then why those scenes are again reconstructed?
> If these videos can be different than the regions covered by the polycam, then how the new poses for the videos are obtained?
>
> The relation between the GT 3D entities, videos, generalisation and assumptions needs to be bit clear for the pipeline to be applied in the novel set of videos where we dont have the GT polycam, and only have the videos over time.
>
> Secondly, the re-encoding of the scene for egocentric localisation can be made a bit simple. Encoding the same scene in two different way may lead to computationally inefficient. Is there way the explicit scene geometry can be leveraged rather than implicit?
>
> Also, SfM/SLAM methods can create the 3D in different coordinate for different videos. But if they are of same scene, they can be registered through rigid transformation. What will be the difficulty in this w.r.t line 208-209

---

> > ### Author Response · Authors · 2024-11-23
> >
> > We appreciate the additional questions and are happy to provide clarifications and update the paper in the [revision](https://openreview.net/pdf?id=y80D4IojuY) accordingly.
> >
> > Q10
> > > If we know the M videos are all corresponds to the 3D map generated using Polycam then why those scenes are again reconstructed? If these videos can be different than the regions covered by the polycam, then how the new poses for the videos are obtained?
> >
> > The scenes in the M videos need to be reconstructed again due to the changes of scenes structure and appearance in the target videos. Over the recording time horizon (e.g., days to months), there exist both global layout changes (e.g., furniture get rearranged) and appearance changes (e.g., table cloth gets replaced), which makes the scene in the target videos different from each other, and also different from the template 3D map.
> >
> > Q11
> > > The relation between the GT 3D entities, videos, generalisation and assumptions needs to be bit clear for the pipeline to be applied in the novel set of videos where we dont have the GT polycam, and only have the videos over time.
> >
> > To train the neural localizer, we assume a template 3D map of the scene. The 3D map is created using Polycam from a single walk-through video with a **complete** capture of the **same** scene as the target videos. To improve generalization ability of the neural localizer on target videos, we randomly sample camera viewpoints and apply image augmentations, including color jitter and masks during training. We have clarified in **L221-230** of the latest revision.
> >
> > Q12
> > > Secondly, the re-encoding of the scene for egocentric localisation can be made a bit simple. Encoding the same scene in two different way may lead to computationally inefficient. Is there way the explicit scene geometry can be leveraged rather than implicit?
> >
> > Thanks for the suggestion of a more efficient way for scene encoding. As we use inverse rendering to reconstruct the agent and scene, a representation that can **faithfully re-create** the pixel measurements in the target videos is preferred. For example, to handle partial occlusion (e.g., a cat being partially occluded by the cushion), the representation needs to be able to represent the cushion to explain the image evidence. Therefore, re-using the explicit 3D scene from Polycam would not be ideal since the geometry and appearance of the scene could change between the template video and the target videos.
> >
> > Q13
> > > Also, SfM/SLAM methods can create the 3D in different coordinate for different videos. But if they are of same scene, they can be registered through rigid transformation. What will be the difficulty in this w.r.t line 208-209
> >
> > Methods for scene alignment (e.g., Procrustes analysis, ICP) with rigid transformation assume stationary points (i.e., the scene does not change) and accurate correspondence. In our case, the scene changes from video to video, and it is difficult to find correspondence due to appearance changes and camera viewpoint differences across videos. We tried rigid ICP and HLoc (Sarlin et al, 2019) and they do not work well due to these two assumptions. We have updated **L202-210** in the revision to make the challenges in multivideo registration clearer.
> >
> > We hope this addresses your concerns and we are open to further questions or aspects needing clarification.

---

> ### Comment · Reviewer_C85q · 2024-11-25
> **Thanks for further clarification**
>
> Now few things are getting cleared.
>
>
> However, if you use the 3D map generated using Polycam as an initial structure and use that to modify based on scene structure then the 3D change along with previous static 3D structure registration has to be detailed in Canonocal structure section. Right now equation 1 and 2 says that the entire structure is trained using MLP from video. The initial 3D is not reflected there and also how to change that 3D based on M videos is also not detailed. This is a very important step and initial 3D assumption in the entire methodology. This will help to understand Q12 above also in a more detailed way..
>
> If you have the 3D scene with the changes, it will be good to comment why resection methods wont work for pose estimation after making the 3D scene complete with the new changes. This will scientifically ground the equation 5.

---

> ### Author Response · Authors · 2024-11-25
>
> Thanks for the additional comments and we hope the following response addresses them.
>
> Q14.
> > However, if you use the 3D map generated using Polycam as an initial structure and use that to modify based on scene structure then the 3D change along with previous static 3D structure registration has to be detailed in Canonocal structure section. Right now equation 1 and 2 says that the entire structure is trained using MLP from video. The initial 3D is not reflected there and also how to change that 3D based on M videos is also not detailed. This is a very important step and initial 3D assumption in the entire methodology. This will help to understand Q12 above also in a more detailed way..
>
> To clarify, the canonical structure is not initialized from the template 3D map. To train the canonical structure, we follow a typical [NeRF](https://arxiv.org/abs/2003.08934) training setting, where the MLPs are initialized with random weights, and optimized by fitting to the image observations using differentiable rendering. In our case, the MLP weights are optimized together with camera poses (fine-alignment), which are initialized from the neural pose regressor (coarse-alignment).
>
> To avoid confusion, we have clarified the initialization of MLPs in **L181** and strengthened the connection to NeRF in **L172** and **L197-198**.
>
>
>
> Q15.
> > If you have the 3D scene with the changes, it will be good to comment why resection methods wont work for pose estimation after making the 3D scene complete with the new changes. This will scientifically ground the equation 5.
>
> Combining the changes in the new videos and the template 3D map needs registration. However, registration is challenging due to non-stationary points and correspondence, as discussed in the previous response. Our method solves registration problem by training a scene-specific neural localizer that bypasses correspondence and rigidity assumption, and provides a good camera initialization for fine-registration (via inverse rendering).
>
> We have updated **L206-214** to make the motivation clearer.
>
> We are happy to answer any further question or aspect that needs clarification.

---

> > ### Comment · Reviewer_C85q · 2024-11-26
> > **Continue to Q14**
> >
> > I understand the response from Q14. But that is also my precise doubt.
> > Why template 3D map cannot be used when it is available? and use of M videos to incorporate the changes. This will be helpful for the reader to understand the motivation of not using that when available. Isnt make easy for the method to get the representation?
> > A detailed discussion may help

---

> ### Author Response · Authors · 2024-11-26
>
> We appreciate the discussion on how to leverage the template.
>
> Q16
> > I understand the response from Q14. But that is also my precise doubt. Why template 3D map cannot be used when it is available? and use of M videos to incorporate the changes. This will be helpful for the reader to understand the motivation of not using that when available. Isnt make easy for the method to get the representation? A detailed discussion may help
>
> Using a template 3D map for initialization is a good idea when there is little changes in the scene. When there are noticeable scene changes, starting from a template has the risk of overfitting, where it is more difficult to revert the changes and fit the new content (as the MLP has converged), compared to starting from scratch.
>
> Please let us know if you have additional comments.

---

> > ### Comment · Reviewer_C85q · 2024-11-27
> > **Thanks, score elevated**
> >
> > I am elevating the score however the following concerns remain and hence the elevation is not significant.
> >
> > 1. From the video it is not clear how much the scene is changed and how the use 3D template and not use of 3D template is affecting the method.
> > 2. The same holds true for the registration and hence this point is still quite unanswerd.
> > 3. The same is valid to understand Q12.
> > 4.Nerf needed the camera pose and that anyway needs to be given so why not start with Polycam output.

---

> ### Author Response · Authors · 2024-11-27
>
> Thanks for raising the score. We try to address the remaining concerns.
>
> To summarize the discussion, the reviewer's question is *"Does having a 3D scene template good enough for **registration** and **reconstruction** of 3D structures from multiple unposed videos?"* We list our understanding of the alternative pipeline being suggested below,
> 1. Pose the template and target videos using off-the-shelf tools (e.g., Polycam), and extract 3D structures.
> 2. **Registration**. Register the target 3D structures to the template 3D structure.
> 3. **Reconstruction**. Use the template 3D structure to initialize NeRF and train it on all videos.
>
> In this pipeline, step 2 (registration) is challenging for the following reasons:
> 1. Geometric registration methods, such as ICP, need good initialization. Since the start location of video recordings are different, the 3D structures are initially largely misaligned, which causes ICP to fail.
> 2. Feature-based registration methods, such as HLoc, requires good visual correspondence, and finding visual correspondence is hard due to two reasons.
>     - (1) Lack of view overlap between videos, e.g., the template video captures the front view of the sofa, but the target videos contain the side view of the sofa.
>     - (2) Changes in appearance across videos, (e.g., lighting, table cloth gets replaced) and layout (e.g., furniture get rearranged).
>
> We experimented with ICP and HLoc and both produced gross errors. Our method trains a neural localizer to provide initialization for registration, which leads to good registration and reconstruction results (Tab. 1, Tab. 2). We are happy to try and compare if the reviewer has alternative methods in mind for step 2.
>
>
> To answer Q4, which is related to step 3 (reconstruction),
> > 4.Nerf needed the camera pose and that anyway needs to be given so why not start with Polycam output.
>
> Although we don't directly use the 3D template to initialize the scene structure, the posed template video is used for reconstruction. We use both the **template video** and new videos to train the scene NeRF. The camera pose of the template video is set to ground-truth cameras from Polycam, and the camera poses of the new videos are initialized from the neural pose regressor and optimized jointly with the scene NeRF. This leverages the accurate pose from Polycam, and avoids overfitting to the template scene.
>
> We have revised the manuscript **L207-214** and **L250-253** to highlight this.

---

### Official Review · Reviewer_Z84r · 2024-11-06

**Soundness:** 2
**Presentation:** 3
**Contribution:** 3
**Rating:** 6
**Confidence:** 4

**Summary:**

This paper proposes Agent-To-Sim (ATS), a framework for learning models of agent behavior conditioned on the observer (ego-perspective) and the static background scene. The scene and the agent are represented using neural radiance field in a canonical state. Additionally, the dynamic motion of the observer and the agent as well as the agent's body deformation is represented using a sequence of SE(3) poses. The latter is achieved by representing the motion as a set of 3D Gaussians in the agent frame and mapping locations by blend-skinning. The NeRF representations are combined and rendered in the respective poses using ray integration. The dynamic scene reconstruction is obtained from video sequences using bundle-adjustment like optimization, initialized with a trained camera pose regressor. Behavior prediction for the agent is trained using a 3-stage diffusion model which subsequently predicts the goal sequence, the path of the agent, and finally its body pose sequence. The approach is evaluated for camera registration, 4D reconstruction, and behavior prediction accuracy quantitatively on one test video sequence for a cat. It is also compared with previous methods as baselines.

**Strengths:**

- The proposed approach to combine 4D scene reconstruction and training diffusion models for behavior prediction seems novel and is interesting.
- The paper is mostly well written and is easy to follow.
- The paper presents two contributions: 1. 4D scene reconstruction including body pose estimation of an agent using NeRF-based representations. 2. A hierarchical diffusion model for behavior prediction in the 4D scene representation.
- The related work section provides a compact overview of the state-of-the-art for related fields.
- The experimental evaluation demonstrates improvements over previous methods for 4D scene reconstruction and agent motion prediction. It also demonstrates improvements over several ablations of the method.

**Weaknesses:**

- The paper is too packed by presenting two contributions at once and by this important details (see below comments) are missing which hinder reproducibility of the approach.
- l. 199ff, it is unclear how the rendering works. How are the NeRF representations of scene and agent combined in detail? Please provide further information either in main paper or supplemental material.
- The paper references directly to several Figures in the appendix which is attached to the main paper without noting that this is supplemental material. Please indicate clearly in the main paper text that they are supplemental material when referencing the figure. Currently, it seems the paper would try to circumvent the page limit.
- What is the orientation representation for regression in Eq. 5 ?
- It is not clear why there is a need for a joint optimization of poses and representations T and D as indicated in Eq 6, because the video sequence is annotated with camera poses from a different (unspecified) system (l. 225). Please clarify which other method is used to obtain the camera motion trajectory. Why is it necessary to optimize the poses in Eq. 6 ? How would the approach perform without this post-optimization? Please add an ablation study. What is the point of training a neural localizer if the camera poses are known from a different system?
- How are the agent root and body part poses intiialized for Eq. 6?
- How are agent and static scene segmented?
- l. 251ff, Is the training and swapping of beta performed across various scenes (i.e. different types of rooms) or only within specific scenes?
- l. 263, why was T*=5.6s chosen and how does the performance of the method depend on the horizon length? This should be evaluated as supplemental results.
- Fig. 2, it is not clear what the thin black arrows mean (e.g. between the motion fields and between score map and goal/path). What is the motion field depicting? Where is this output in the proposed method?
- l. 304ff., the note on generalization to different actions seems problematic. The datasets used in the experiments seem small. The biggest dataset (cat) has only 22 training videos of a few minutes each. The anecdotal result pointed to in the supplemental material (Fig. 11) might be a rare example. Please discuss and revise the claims.
- Eq. 11, what is \Psi_s ? Please define.
- l. 314ff, the text mentions trajectories, but the equation only transforms a single pose (\xi^a). Please clarify.
- l. 318ff, please explain why the approach uses the observer trajectory perceived by the agent and not the other way round. Please also provide an evaluation if this makes a difference.
- l. 360, a segmentation method is mentioned,but it is not stated how it is used.
- l. 401. GT relative camera pose is obtained by some unspecified process. How does it work in detail and how accurate is this process? Please provide this description in the supplemental material.
- l. 428ff / Tables 4 and 5: the motion prediction is conditioned on ground-truth goal and path for the evaluation. However, this should also be evaluated for regressed goal and path to fully evaluate the proposed method without such oracle information.
- How were ground-truth goal and path annotated in the data ?
- It seems the paper lists several datasets (bunny, human, etc), but only shows results for the cat videos.
- If possible, the datasets used in the evaluation should be made publicly available for future comparison with the proposed method. Is it planned to release the datasets publicly ? Please also specify how this data was collected. Are there potential personal data protection issues for releasing the data?
- The datasets used for evaluation are rather small and, hence, results seem anecdotal and significance is unclear. The method should be evaluated quantitatively on more and longer sequences and a larger variety of agents (e.g., 10-20 different animals and human subjects) and scenes (e.g. 10-20 different rooms) to increase significance of results and to provide further insights into the performance and limitations of the proposed method. Also the variety of daily activities should be increased and annotated. Cultural and gender diversity of subjects should also be considered when designing a benchmark dataset. With larger datasets, also the dependence of the performance of the method on the amount of available training data could be evaluated.

Further minor comments:
- p. 2, please add a label/caption to the lower figure and reference it from the text.
- l. 76ff, the notation for \sigma, c, \psi and the variants c_s/a, \sigma_s/a, and \psi_s/a is not accurate. Are the subscripts referencing parts of the variables without subscript? What kind of mathematical structure are they?
- l. 210 "buil" => "build"
- Fig. 2 caption "fully" => "full"
- Eq. 11, "X_w" should be "X^w"
- l. 369, the reference to Eq 5 should point to Eq 6

**Questions:**

- Please address the questions raised in paper weaknesses in the rebuttal. Please address the lacking details of the method description and the raised shortcomings of the evaluation.

---

> ### Author Response · Authors · 2024-11-22
>
> Thanks for the encouraging comments and detailed feedback. We have revised the paper to address the comments and provided additional details for reproducibility. Key text changes are highlighted in **dark red** in the [revision](https://openreview.net/pdf?id=y80D4IojuY).
>
> Below, we provide answers to the questions raised and would welcome any further feedback or clarification requests.
>
> Q1
> > How are the NeRF representations of scene and agent combined in detail?
>
> We revised **L193-199** and provided additional details on compositional rendering in **Appendix A.1**.
>
> Q2
> > Reference of figures in the supplement.
>
> We have clarified that the figures are part of the appendix in the revision.
>
> Q3
> > The orientation representation for regression in Eq. 5.
>
> We use unit quaternion to represent rotations. We force the real part to be positive to avoid the ambiguity in the representation, as quaternions q and -q represent the same rotation. This has been clarified in **L228-229** of the revision.
>
> Q4
> > It is not clear why there is a need for a joint optimization of poses and representations T and D as indicated in Eq 6, because the video sequence is annotated with camera poses from a different (unspecified) system (l. 225). Please clarify which other method is used to obtain the camera motion trajectory. Why is it necessary to optimize the poses in Eq. 6 ? How would the approach perform without this post-optimization? Please add an ablation study. What is the point of training a neural localizer if the camera poses are known from a different system?
>
> Although one could obtain camera trajectory in a **single video** using off-the-shelf SfM/SLAM tools, when optimizing **multiple** videos of the same scene, the per-video trajectories are not globally aligned to the same coordinate system. The joint optimization aligns camera poses from all videos to a global scene coordinate system. Without Eq. 6, our method degrades to Multivideo-TotalRecon and produces worse reconstruction results (DepthAcc: 0.708 vs 0.099), as shown in Tab. 3. We have clarified the motivation of multi-video joint optimization in **L202-210** of the revision.
>
> Q5
> > How are the agent root and body part poses initialized for Eq. 6?
>
> The body part poses are initialized as identity and the root pose is initialized from pose estimator. We have clarified in **L245-246** of the revision.
>
> Q6
> > How are agent and static scenes segmented?
>
> We use Grounding DINO to annotate a bounding box given text description of the agent (e.g., cat), and SAM to segment the agent in the first frame of the video. The segmentation is tracked over all the frames using XMem. We took the existing code from Lab4d, which is an open-source project.  This has been clarified in **L845-849** of the revision.
>
> Q7
> > Is the training and swapping of beta performed across various scenes (i.e. different types of rooms) or only within specific scenes?
>
> The training and swapping is performed for the same scene across multiple videos and days.
>
> Q8
> > l. 263, why was T*=5.6s chosen and how does the performance of the method depend on the horizon length? This should be evaluated as supplemental results.
>
> We evaluated the behavior prediction with different prediction horizons T*={4.0, 5.6, 7.2}s, and results are updated to **Tab. 4** and **L482-483** of the revision. We found the longer the prediction horizon is, the more difficult to predict the goals and future paths. T*=5.6s was chosen following prior works (Rempe et al. 2023), which strikes a balance between the generation time and the coherence of the generated content.
>
> Q9
> > Fig. 2, it is not clear what the thin black arrows mean (e.g. between the motion fields and between score map and goal/path). What is the motion field depicting? Where is this output in the proposed method?
>
> Black arrows in Fig. 2. visualize the output of the denoising network (Eq. 7), which points to the direction to update the goal locations in the iterative denoising process. This has been clarified in **L284-285** of the revision.
>
> Q10
> > l. 304ff., the note on generalization to different actions seems problematic. The datasets used in the experiments seem small. The biggest dataset (cat) has only 22 training videos of a few minutes each. The anecdotal result pointed to in the supplemental material (Fig. 11) might be a rare example. Please discuss and revise the claims.
>
> Thanks for pointing it out. We acknowledge that the datasets used in our experiments are relatively small, and we understand your concern that the anecdotal result in Figure 11 may not provide sufficient evidence for broad generalization. We have revised the claim about generalization from our paper. Instead of stating “We find that a specific behavior can be learned and generalized to novel situations even when seen once”, we revise it to “This enables the model to generate novel scenarios that were not present in the training dataset.” in **L308-309**.

---

> ### Author Response · Authors · 2024-11-22
>
> Q11
> > Eq. 11, what is \Psi_s ? Please define.
>
> $\Psi_s$ is the scene feature queried at point $X_w$ through= Eq. 1. We have clarified it in the revision **L314-315**.
>
> Q12
> > l. 314ff, the text mentions trajectories, but the equation only transforms a single pose (\xi^a). Please clarify.
>
> Thanks for pointing it out. This has been revised in **L320-353**.
>
> Q13
> > l. 318ff, please explain why the approach uses the observer trajectory perceived by the agent and not the other way round. Please also provide an evaluation if this makes a difference.
>
> The alternative way of encoding the state of the agent with respect to the observer is equivalent to our formulation when there is only one “observer/external agent”. However, this alternative formation is not extendible to more than one observer (external agent), while our formulation could be more intuitive when handling more than one external agents.
>
> Q14
> > l. 360, a segmentation method is mentioned,but it is not stated how it is used.
>
> We use the segmentation as the ground-truth for object silhouette loss in **L249**, which is necessary to separate the agent from the background. We updated the text from "silhouette loss" to "segmentation loss" to make it clearer.
>
> Q15
> > l. 401. GT relative camera pose is obtained by some unspecified process. How does it work in detail and how accurate is this process? Please provide this description in the supplemental material.
>
> We have updated the annotation process of GT camera pose in the appendix **L863-902**. The accuracy of camera pose annotation depends on the precision of 2D keypoint annotation and accuracy of the depth map. Solving relative camera poses has an exact solution when both measurements are perfect an d the point pairs are not degenerate.
>
> Q16
> > l. 428ff / Tables 4 and 5: the motion prediction is conditioned on ground-truth goal and path for the evaluation. However, this should also be evaluated for regressed goal and path to fully evaluate the proposed method without such oracle information. How were ground-truth goal and path annotated in the data ?
>
> Thanks for catching this. We made a mistake in writing and we’ve updated the text. In Tab. 4 "End-to-end Evaluation of Interactive Behavior Prediction", the predicted behavior is not conditioned on oracle information (GT goal or path). In Table 5 "Spatial Control", the path generation is conditioned on GT goal, and the full body motion generation is conditioned on GT path. The GT goal is the position of the agent’s root T*=5.6s into the future, and GT path is defined as the trajectory of the agent’s root T* seconds into the future.

---

> ### Author Response · Authors · 2024-11-22
>
> Q17
> > It seems the paper lists several datasets (bunny, human, etc), but only shows results for the cat videos.
>
> We show visual results of cat, bunny, human, and dog on the supplementary webpage. It includes 4D reconstruction for all video sequences of all agents, and behavior generation results for cat and human.
>
> Q18
> > If possible, the datasets used in the evaluation should be made publicly available for future comparison with the proposed method. Is it planned to release the datasets publicly ? Please also specify how this data was collected. Are there potential personal data protection issues for releasing the data?
>
> The data is collected from the friends of the researchers that contributes to the project, and the dataset will be made publicly available upon paper acceptance.
>
> Q19
> > The datasets used for evaluation are rather small and, hence, results seem anecdotal and significance is unclear. The method should be evaluated quantitatively on more and longer sequences and a larger variety of agents (e.g., 10-20 different animals and human subjects) and scenes (e.g. 10-20 different rooms) to increase significance of results and to provide further insights into the performance and limitations of the proposed method. Also the variety of daily activities should be increased and annotated. Cultural and gender diversity of subjects should also be considered when designing a benchmark dataset. With larger datasets, also the dependence of the performance of the method on the amount of available training data could be evaluated.
>
> We appreciate the valuable feedback regarding the scope and scale of the datasets used for evaluation. While we acknowledge that larger and more diverse datasets could provide additional insights and strengthen the evaluation, we would like to clarify the focus and constraints of our work. Our goal was to study this understudied problem of learning behaviors of a single individual from long observations. While there are studies that collects large-scale behavior datasets, learning from a videos of the same agent is less studied and we are in this camp.
>
> We agree that creating a comprehensive benchmark dataset with diverse subjects, activities, and environments would be valuable. However, constructing such a dataset at the scale suggested is nontrivial, particularly around privacy concerns. While this is outside the immediate scope of this work, we believe that open-sourcing our code will enable the community to build upon our framework and contribute to the development of larger, more diverse benchmarks collaboratively. We hope this clarifies the intent and scope of our work.
>
> > Minor comments
>
> Thanks for the feedback! Those have been fixed in the revision.
>
> We hope these revisions and explanations address the concerns raised. Please let us know if there are further questions or aspects needing clarification.

---

> ### Comment · Reviewer_Z84r · 2024-11-25
> **Thanks for author response**
>
> Thanks for the insightful response which addresses several of my concerns well.
>
> Further comments/questions:
>
> - Results which are only provided on a volatile supplemental webpage cannot be used in a published paper to support claims, since the webpage is not provided with the paper by the publisher. Hence, please restrict the claims/result discussion to the datasets used in the main paper (cat; bunny for the specific experiment in Fig. 4).
>
> - How is the ground truth for the agent motion obtained/annotated in the dataset? Please detail in the paper or supplemental material.

---

> > ### Author Response · Authors · 2024-11-25
> >
> > Thank you the additional feedback and please find our response below.
> >
> > Q20
> > > Results which are only provided on a volatile supplemental webpage cannot be used in a published paper to support claims, since the webpage is not provided with the paper by the publisher. Hence, please restrict the claims/result discussion to the datasets used in the main paper (cat; bunny for the specific experiment in Fig. 4).
> >
> > Thanks for the suggestion. We have removed the claims about dog and human in the main paper, and mentioned the results on human and dog in the appendix.
> >
> > Q21
> > > How is the ground truth for the agent motion obtained/annotated in the dataset? Please detail in the paper or supplemental material.
> >
> > We have revised the manuscript to include more details about agent motion data in **L406-411**.
> >
> > We hope our response addresses the questions. We are happy to answer further questions and take additional feedback.

---

### Official Review · Reviewer_kcfP · 2024-11-06

**Soundness:** 3
**Presentation:** 4
**Contribution:** 3
**Rating:** 8
**Confidence:** 5

**Summary:**

This paper proposes a framework called Agent-to-Sim to learn the interactive behaviors of 3D agents in a 3D environment from casually captured videos. Specifically, a coarse-to-fine registration method is developed for persistent and complete 4D representations. A generative model of agent behaviors of agent behaviors is trained to enable the generation of agent's reactions to the observer's motions. Interesting demos show the effectiveness of the proposed system.

**Strengths:**

1. This paper proposes a framework to learn interactive behaviors of agents in 3D worlds, which I believe is an interesting and promising direction. With the rise of embodied AI, this paper offers a novel approach to leveraging real-world videos for understanding interactions between agents and their environment.

2. This paper is clearly written and well-presented, with an impressive demo that effectively showcases the benefits of the proposed framework.

3. The proposed methods are logical for building the entire system and have been thoroughly validated through experiments.

**Weaknesses:**

1. In terms of behavior generation, there is extensive existing work on generating realistic human motions based on intentions or trajectories, with most approaches utilizing parametric models like SMPL. Given that the agent in the demo appears somewhat unrealistic, I was curious why parametric models weren't used for agent representation. For example, there are parametric models available for animals, such as [1,2].

[1] Zuffi et al.: 3D Menagerie: Modeling the 3D Shape and Pose of Animals, CVPR 2017

[2] Zuffi et al.: Lions and Tigers and Bears: Capturing Non-Rigid, 3D, Articulated Shape from Images, CVPR 2018

2. Regarding the egocentric perception: is the egocentric visual/video data encoded as well? It is mentioned "a latent representation from a
local feature volume around the agent", but an agent like a cat can not see the scene behind it. Besides, the proposed egocentric encoding to transform the world to the egocentric coordinates to avoid over-fitting as well as past encoding to capture the past motion sequence seem to have been proposed and well discussed in [3], which also predicts the motions from spatial controls.

[3] Jiang et al.: EgoPoser: Robust Real-Time Egocentric Pose Estimation from Sparse and Intermittent Observations Everywhere, ECCV 2024

3. The proposed system is trained and tested on the collected dataset. However, based on the information provided in the paper, this dataset appears quite small and lacks diversity, as it includes only 4 agents, 3 scenes, and has a limited duration. Given the availability of numerous large datasets featuring moving agents and varied environments, incorporating these existing datasets for evaluation would strengthen the findings. Additionally, I am curious about the generalization capability of the proposed method, as the experiments were conducted by training on only 22 videos and testing on a single remaining video. Do the training and testing data contain the same scenes? If so, this might lead to potential overfitting issues.

4. I am interested in the potential applications of the proposed systems beyond the general areas mentioned in the paper, such as driving, gaming, and movies. It would be valuable to explore specific existing tasks to demonstrate how the system can provide practical benefits.

**Questions:**

See above

---

> ### Author Response · Authors · 2024-11-22
>
> Thanks for the encouraging comments! We have revised the paper to address the feedback. Key text changes are highlighted in **dark red** in the [revision](https://openreview.net/pdf?id=y80D4IojuY).
>
> Below, we provide answers to the questions raised and would welcome any further feedback or clarification requests.
>
> Q1
> > Given that the agent in the demo appears somewhat unrealistic, I was curious why parametric models weren't used for agent representation. For example, there are parametric models available for animals, such as [1,2].
>
> While parametric body models have their advantages, they also limit the type of animals that can be represented in nature. For example, bunnies and mice are outside SMAL family. To demonstrate that our approach is applicable to a wider range of animals, we use BANMo (NeRF+articulated bones) since it is flexible to explain more animal categories. We have also revised the paper and discussed [1,2] in the related works **L113-118**.
>
> Q2
> > Regarding the egocentric perception: is the egocentric visual/video data encoded as well? It is mentioned "a latent representation from a local feature volume around the agent", but an agent like a cat can not see the scene behind it.
>
> We do not record/use egocentric RGB videos. The iPhone (third-person view) videos are lifted to 3D (Eq. 1-2) and encoded as the scene input to the behavior prediction model. The latent representation of the scene in Eq. (11) is a 3D convolution of the DINO-v2 feature volume of the room in the agent's perspective. This encoding mostly captures the immediate surroundings of the agent. Encoding the visible surroundings might be a better design choice, but it is also not perfect due to the lack of other sensor modalities of the agents, such as audio and touch. We leave this for future investigation.
>
> Q3
> > The proposed egocentric encoding to transform the world to the egocentric coordinates to avoid overfitting as well as past encoding to capture the past motion sequence seem to have been proposed and well discussed in [3], which also predicts the motions from spatial controls.
>
> Please note that our model predicts future behavior of an agent from synthesized egocentric perception, as opposed to works like [3] which predict the full body motion from head-mounted sensor inputs. [3] is also a concurrent work that appears in ECCV 24 (which is after the ICLR submission). We appreciate the pointer and have cited it in the revision **L307-308**.
>
> Q4
> > The proposed system is trained and tested on the collected dataset. However, based on the information provided in the paper, this dataset appears quite small and lacks diversity, as it includes only 4 agents, 3 scenes, and has a limited duration. Given the availability of numerous large datasets featuring moving agents and varied environments, incorporating these existing datasets for evaluation would strengthen the findings. Additionally, I am curious about the generalization capability of the proposed method, as the experiments were conducted by training on only 22 videos and testing on a single remaining video. Do the training and testing data contain the same scenes? If so, this might lead to potential overfitting issues.
>
> We appreciate the valuable feedback regarding the scale of the datasets used for evaluation. While we acknowledge that using a larger and more diverse datasets could provide additional insights and strengthen the evaluation, we would like to clarify the focus of our work. We are exploring a novel problem setup, where video observations of the same agent across a long time horizon (weeks) is available. The goal of the model is to **overfit** to a single agent and scene. Therefore, we only study generalization to **different time instances**. To show the generality of the algorithm, we experiment our approach on a wide variety of agents: cat, human, dog, bunny.
>
> Q5
> > I am interested in the potential applications of the proposed systems beyond the general areas mentioned in the paper, such as driving, gaming, and movies. It would be valuable to explore specific existing tasks to demonstrate how the system can provide practical benefits.
>
> Thanks for the comment. We also hope the methods proposed in this paper will be useful for generating plausible behaviors for characters in gaming and interactive movies, as well as for embodied agents in the physical world. We leave this for future investigation.
>
> We hope these revisions and explanations address the concerns raised. Please let us know if there are further questions or aspects needing clarification.

---

> > ### Comment · Reviewer_kcfP · 2024-11-25
> >
> > Thanks for the rebuttal. I appreciate the author's response, and I have also read the revised paper. Most of my concerns were addressed. So I decide to increase my score from 6 to 8.

---

### Meta-Review · Area_Chair_xTre · 2024-12-18

**Metareview:**

This paper proposes a method called Agent-To-Sim (ATS) for learning interactive behavior models from casual videos. Specifically, a 4D representation is reconstructed from the input videos, consisting of time-independent 3D canonical Structures and time-varying components. In order to align multiple videos to a global world coordinate, the paper proposes a coarse-to-fine multi-video registration approach. After the 4D representation is obtained, three diffusion models are trained to respectively generate goals, paths and body poses conditioned on ego-perception.

The main strengths of this paper are as below:
- This paper introduces a framework for learning the interactive behaviors of agents from casually-captured videos, which is an interesting and promising direction.
- This paper makes two contributions: (1) A coarse-to-fine multi-video registration approach for reconstructing 4D scene representations from multiple videos. (2) A hierarchical diffusion model for behavior generation in 4D scene representations.
- The experimental evaluation demonstrates improvements over previous methods for multi-video 4D scene reconstruction and interactive behavior generation.

Needing more experiments and analyses, improving the writing  were raised by the reviewers. Please revise the paper according to the discussions before submitting the final version.

**Additional Comments On Reviewer Discussion:**

- Reviewer kcfp asked for the reason why the paper utilizes NeRF-based model BANMo instead of parameter models like SMPL to represent agents. The authors explained the limitations of those parameter models like SMPL, which are restricted in representing a limited range of animal types, whereas the BANMo used in this paper offers more flexibility in representing a broader range of animal categories. Reviewer kcfp raised that the dataset used in this paper includes a limited number of animal categories, which restricts the generalizability of the proposed method. The authors emphasized that their research focuses on generalization across different time instances rather than across animal categories.
- Reviewer Z84r raised that the paper is too packed and by this important details are missing. The authors provided explanations to address the lacking details in the paper. Reviewer Z84r pointed out some writing errors in the paper. The authors made the necessary corrections accordingly.
- Reviewer C85q pointed out that the writing in this paper is rather complex and difficult to understand, and requested the authors to provide more detailed explanations of the technical details used in the article. The authors have accordingly provided explanations and supplementary details on the technical aspects of the article, and have updated them in the revised manuscript.
- Reviewer 8uBB asked about how to obtain training data for neural camera localization. The authors responded that the training data is generated from 3D maps. Reviewer 8uBB asked about more specific details in the paper. The authors provided corresponding responses to all the queries.

---

### Decision · Program_Chairs · 2025-01-22

Accept (Poster)